# 3-D Cloud Masking Across a Broad Swath using Multi-angle Polarimetry and Deep Learning

Sean R. Foley[1,2], Kirk D. Knobelspiesse[1], Andrew M. Sayer[1,3], Meng Gao[1,4], James Hays[5], and Judy Hoffman[5]

[1]NASA Goddard Space Flight Center, Code 616, Greenbelt, MD 20771, USA
[2]Goddard Earth Sciences Technology and Research (GESTAR) II, Morgan State University, Baltimore, MD, USA
[3]Goddard Earth Sciences Technology and Research (GESTAR) II, University of Maryland, Baltimore County, Baltimore, MD, USA
[4]Science Systems and Applications, Inc., Greenbelt, MD, USA
[5]Georgia Institute of Technology, Atlanta, GA, USA

**Correspondence:** Sean Foley (sean.r.foley@nasa.gov)

**Abstract.** Understanding the 3-dimensional structure of clouds is of crucial importance to modeling our changing climate. Both active and passive sensors are restricted to two dimensions: as a cross-section in the active case, and an image in the passive case. However, multi-angle sensor configurations contain implicit information about 3D structure, due to parallax and atmospheric path differences. Extracting that implicit information requires computationally expensive radiative transfer techniques. Machine learning, as an alternative, may be able to capture some of the complexity of a full 3D radiative transfer solution with significantly less computational expense. In this work, we develop a machine learning model that predicts radar-based vertical cloud profiles from multi-angle polarimetric imagery. Notably, these models are trained only on center-swath labels, but can predict cloud profiles over the entire passive imagery swath. We compare with strong baselines and leverage the information-theoretic nature of machine learning to draw conclusions about the relative utility of various sensor configurations, including spectral channels, viewing angles, and polarimetry. Our experiments show that multi-angle sensors can recover surprisingly accurate vertical cloud profiles, with skill strongly related to the number of viewing angles and spectral channels, with more angles yielding high performance, and the oxygen A-band strongly influencing skill. A relatively simple convolutional neural network shows nearly identical performance to the much more complicated U-Net architecture. The model also demonstrates relatively lower skill for multilayer clouds, horizontally small clouds, and low-altitude clouds over land, while being surprisingly accurate for tall cloud systems. These findings have promising implications for the utility of multi-angle sensors on Earth-observing systems such as NASA's Plankton, Aerosol, Cloud-ocean Ecosystem (PACE) and Atmosphere Observing System (AOS), and encourage future applications of computer vision to atmospheric remote sensing.

## 1 Introduction

Clouds regulate the global climate system in many important ways. Cloud radiative effects can be both warming and cooling, depending on the type and altitude of the cloud (Stephens and Webster, 1981). As the climate changes, the global cloud distribution will be affected by numerous feedback cycles, some of which are not fully understood (Ceppi et al., 2017; Gettelman and

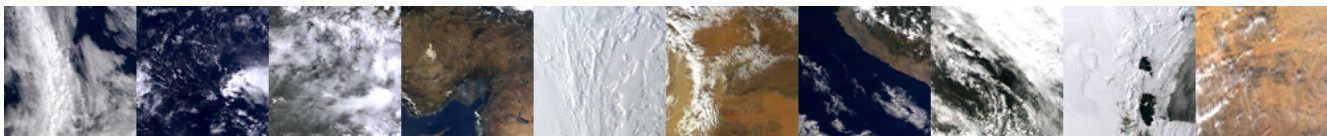

**Figure 1.** True color POLDER-3 images from the test set, illustrating a diversity of cloudy scenes.

Sherwood, 2016). As a result, clouds and cloud feedbacks are among the greatest sources of uncertainty in climate sensitivity models (Pörtner et al., 2022; Meehl et al., 2020; Bony et al., 2015). Monitoring the response of global cloud distributions to a changing climate is of utmost importance in the coming decades. Understanding the 3D structure of clouds, as well as the ver-
tical distribution of cloud phase and thickness, is of great importance to climate modeling studies, such as the characterization of positive and negative cloud feedbacks (Rossow et al., 2022; Marchand et al., 2010).

Satellite remote sensing can help reduce these uncertainties by providing a source of consistent, global data. Three satellite missions are of particular relevance to this work. One of these is Polarization and Anisotropy of Reflectances for Atmospheric Sciences coupled with Observations from a Lidar (PARASOL), a French Centre National D'études Spatiales (CNES) mission
which launched in 2004 and remained operational until 2013. POLarization and Directionality of the Earth's Reflectances (POLDER) was a multi-angle polarimeter, the third of which (POLDER-3) was mounted on PARASOL. It was designed to improve understanding of clouds, aerosols, and their climate interactions (Deschamps et al., 1994; Buriez et al., 1997). POLDER data continues to provide diverse insights on aerosol properties near clouds (Waquet et al., 2013), constraints on global emissions (Chen et al., 2019), multilayer cloud identification (Desmons et al., 2017), and more. The second relevant
mission is CloudSat - a NASA mission which observed the vertical distribution of clouds, aerosols, and precipitation using backscatter from its Cloud-Profiling Radar (CPR) (Stephens et al., 2002; Im et al., 2005). From its launch in 2006 to its end of life in 2023, CloudSat provided diverse insights on clouds, aerosols, and precipitation (L'Ecuyer et al., 2015; Smalley et al., 2014; Haynes et al., 2009; Liu, 2008). Finally, Cloud-Aerosol Lidar and Infrared Pathfinder Satellite Observations (CALIPSO), which launched in 2006, carried both an active lidar sensor and passive imagers, enabling the study of the vertical distribution
of clouds and aerosols (Winker et al., 2009). All three of these satellites at one point shared an orbital constellation known as the A-Train, yielding nearly-simultaneous observations of the surface (Stephens et al., 2018). CloudSat left the A-Train in 2018, followed by CALIPSO, but remained in orbit for several more years. These missions have reached end of life now, but remain an invaluable resource for preparing for the next generation of atmospheric satellites, like PACE (Werdell et al., 2019).

The characterization of the vertical structure of clouds can be done from both passive and active sensors. Cloud-top height
is often derived from passive thermal infrared measurements (Baum et al., 2012). These tend to provide only the top altitude of clouds in a column and struggle in multilayered cloud systems (Holz et al., 2008; Mitra et al., 2021), but have the advantage of broad spatial coverage. Active sensors, like CloudSat's CPR, have the advantage of providing a more complete vertical profile, at the expense of coverage. For example, CloudSat observed a narrow cross-section of clouds at nadir, whereas POLDER's maximum swath width (across view angles) was approximately 2200 kilometers (Buriez et al., 1997). In addition, some passive
sensors are multi-angle, allowing the use of stereoscopic methods to retrieve cloud structure.

Prior work has explored the usage of A-Train data to understand the structure of clouds, e.g. (Barker et al., 2011), in which off-nadir cloud properties are estimated by nearest neighbor matching with actively observed pixels, using Moderate resolution Imaging Spectroradiometer (MODIS) radiances. Another approach estimates the cloud profiles from MODIS using a conditional generative adversarial network (Leinonen et al., 2019). Another work has explored the usage of POLDER data to understand the structure of clouds, using a decision tree to predict whether each pixel contains single-layer or multilayer clouds (Desmons et al., 2017). Observations from the Multi-angle Imaging SpectroRadiometer (MISR) in tandem with MODIS have been used to estimate two-layer cloud properties (Mitra et al., 2023). Airborne multi-angle platforms like the Research Scanning Polarimeter (RSP) have been used to study multilayer clouds (Sinclair et al., 2017). The estimation of vertical cloud profiles in geostationary data using machine learning has been studied (Brüning et al., 2023). This concurrent work employs a similar strategy to ours, but on geostationary, single-view imagery. To our knowledge, our work is the first to estimate full vertical cloud profiles from POLDER data.

The problem of reconstructing 3D geometry has a rich history in both remote sensing and computer vision, but the techniques used differ greatly between those fields. Whereas computer vision techniques nearly always assume a pinhole camera model, remote sensing uses a variety of different models, depending on the sensor's design. One common model for pushbroom sensors is the rational polynomial camera model (Gupta and Hartley, 1997; Zhang et al., 2019), which is incompatible with standard 3D reconstruction pipelines. Additionally, remote sensing reconstruction typically operates on individual stereo pairs, rather than simultaneously reconstructing more than two views (Schonberger and Frahm, 2016). Even within the remote sensing space, most 3D reconstruction approaches operate on relatively high resolution imagery (on the order of several meters, e.g. Castro et al. (2020)), and not on wide-swath imagery like POLDER, whose pixels are 6x7 kilometers wide (Buriez et al., 1997). Another scale-related difficulty is the difference between horizontal and vertical resolution. Many vertical cloud profiles are reported at sub-kilometer vertical resolution (Stephens et al., 2002), which may be difficult or impossible to accurately retrieve from low-resolution passive imagery like POLDER. Another example of 3D reconstruction is the MISR Interactive eXplorer project (MINX), which performs stereo height retrieval of aerosol plumes in MISR data (Nelson et al., 2010). However, the smaller pixel size and larger parallax simplifies the determination of height. Cloud-top height retrievals were performed with the Along-Track Scanning Radiometer (ATSR) sensor series (Muller et al., 2007). Both MISR and ATSR have sharper resolutions than the $6 \times 7 \text{ km}^2$ pixels of POLDER (Buriez et al., 1997), with MISR at 275-1100 m (Diner et al., 1998), and with ATSR at 1 km (Muller et al., 2007).

As an alternative to stereoscopic 3D reconstruction, we elect to directly estimate the vertical cloud distribution of each location. This is accomplished with a deep learning method, which essentially takes passive imagery as input and estimates an active sensor product as output, but on the same spatial grid as the input. The model takes a flat representation of the multi-angle imagery as an input and produces a dense 3D binary cloud / no-cloud grid. The 3D masking task is known in the computer vision literature as volumetric segmentation (as opposed to image segmentation). Deep learning for volumetric segmentation is well-studied, particularly within medical applications like computed tomography (Çiçek et al., 2016; Soffer et al., 2021; Ardila et al., 2019; Jnawali et al., 2018). However, these methods take a 3D input. By contrast, our method stacks the various viewpoints of the surface (as seen from different angles), as is done in 3DeepCT (Sde-Chen et al., 2021), in which deep learning

**Table 1.** POLDER-3 spectral bands.

| Band (nm) | 443 | 490 | 565 | 670 | 763 | 765 | 865 | 910 | 1020 |
|---|---|---|---|---|---|---|---|---|---|
| Bandwidth (nm) | 20 | 20 | 20 | 20 | 10 | 40 | 40 | 40 | 20 |
| Polarization | no | yes | no | yes | no | no | yes | no | no |

is used to regress 3D liquid water content. Our motivation for stacking multiple viewpoints is "depth-from-disparity", where the disparity (in appearance between viewpoints) is an indicator of the depth (distance from sensor). Another highly relevant work is VIP-CT (Ronen et al., 2022), which directly regresses the extinction coefficient of a 3D cloud field using simulated multi-angle data.

## 2   Data

In order to satisfy the data requirements of a deep neural network, we synthesize POLDER data with a CloudSat product called 2B-CLDCLASS (Sassen and Wang, 2008). Both the original POLDER and CloudSat data are made available by the AERIS/ICARE Data and Services Center (see Code and Data Availability). The original POLDER data comes from the level-1B product, whereas for CloudSat we use a product called 2B-CLDCLASS as provided by the Calxtract application from ICARE. We call the fused and re-formatted dataset the A-Train Cloud Segmentation (ATCS) Dataset, after the orbital constellation these satellites shared. The ATCS dataset and related API are available as an archive in the SeaWiFS Bio-optical Archive and Storage System (SeaBASS).

### 2.1   POLDER data

The POLDER level-1B data used in this study are in HDF5 format, organized by date. Each file corresponds to one PARASOL half-orbit (the daytime side). POLDER level 1 and higher products are generated on a gridded equal-area sinusoidal projection, with each pixel in this grid corresponding to approximately $6 \times 7$ km on the surface (Hagolle et al., 1996). Each pixel contains the co-registered quasi-simultaneous multi-angular viewpoints. There are up to 16 viewing angles (from PARASOL's perspective) from which a point on the Earth can be observed. The maximum delay between these quasi-simultaneous observations is on the order of several minutes. See Table 1 for a description of the POLDER-3 spectral bands.

### 2.2   2B-CLDCLASS

These files are merged with 2B-CLDCLASS data, a CloudSat product which contains CPR data, classified into eight cloud types (Sassen and Wang, 2008). While the eight cloud types preserved in the ATCS dataset, this study treats cloud presence as a binary label. The CPR range extends from beneath the surface up to 25 km, the vertical resolution is 240 m, and there are

125 height bins in the data. There are few to no clouds in many of the higher altitude bins (above 14 km) for the sampled data,
which are discarded. Sub-surface bins are discarded as well. There are 59 bins in the valid range.

There are related products which combine the CloudSat radar and the CALIOP lidar, such as 2B-GEOPROF-LIDAR (Mace and Zhang, 2014). When including lidar, the data contain many optically thin clouds, especially cirrus clouds, to which POLDER and many other passive sensors are often not sensitive. This caused issues when applying supervised learning; the loss function was dominated by optically thin clouds and performance on the radar-observed clouds suffered as a result. This course of study was therefore discontinued, but could prove interesting for future work.

Generally, POLDER-3 and CPR are sensitive to different physics. It should be impossible for a method with access to only the shortwave information in POLDER-3 to fully capture a radar product. Some loss of information is expected, but this study aims to provide a reasonable lower bound on the degree to which POLDER-3 contains the 2B-CLDCLASS information. This goes both ways: MODIS is sensitive to some cloud features that are not detected by CloudSat and CALIOP, particularly low, optically thin clouds (Chan and Comiso, 2011; Christensen et al., 2013). POLDER-3 should also be able to detect these features, given its similar spectral sensitivity to MODIS. We focus our efforts on quantifying the passive-to-radar estimation rather than the converse, in order to capture the benefits of a wide swath.

## 2.3 Sampling Strategy

As the quantity of available data is more than sufficient to train a deep network, we make use of uniform sampling to compile the dataset. We randomly sampled half-orbit files for both PARASOL / POLDER and CloudSat / 2B-CLDCLASS from every day in a predetermined date range, and discarded invalid data. The date range was determined by the availability of valid data: before November 27, 2007, our data extraction process was unable to recover valid data. In December 2009, PARASOL lowered its orbit to exit the A-Train (Stephens et al., 2018), meaning POLDER-3 data were no longer quasi-simultaneous with CloudSat. Data were considered invalid under any of the following conditions: 1) any of the POLDER, 2B-CLDCLASS, or CALIPSO/CALIOP half-orbit files were missing, 2) the half-orbit files start-times were off by more than ten minutes (potentially indicating an incomplete record), 3) the produced records did not contain valid data for 13 or more angles in the POLDER imagery, or 4) the produced records did not contain 50 or more labeled pixels.

Standard practice in machine learning involves splitting datasets into separate sets for training, validation, and testing. Labels from the training set are used to optimize the model's parameters. The validation set is used to examine the model's performance, and its labels are held out so they cannot directly affect the model's parameters. However, hyperparameters (e.g. learning rate, how many layers in a model) are typically optimized with respect to the validation set. Neither parameters nor hyperparameters should be optimized with respect to the test set, which is the final measure of a method's efficacy. Commonly, the training set is the largest. We separately generate a training + validation (trainval) set and a test set, with the test set being approximately one quarter the size of the trainval set. The trainval set is then split with 80% of half-orbit files assigned to training and 20% assigned to validation.

The trainval set has two files per day, and the test set has one file per day. For each 2B-CLDCLASS half-orbit file, we uniformly sampled random locations along the CloudSat track, attempting the alignment procedure described in the next

section until acquiring enough patches: 16 for the trainval set, 8 for the test set. The patches are 100x100 pixels, as these have sufficient spatial context without yielding overly large file sizes. Since each pixel contains hundreds of values corresponding to various angles and spectra, increasing patch size quickly becomes costly to storage space. Conversely, smaller patch sizes reduce spatial context, which contains useful information for the segmentation algorithm.

## 2.4 Alignment Strategy

Alignment between POLDER and 2B-CLDCLASS data is performed with sub-pixel accuracy, with respect to POLDER. The POLDER grid is a sinusoidal (equal-area) projection of the globe (Hagolle et al., 1996). Each pixel in this grid contains multi-angular sensor data, geography (latitude, longitude, altitude), and geometry (viewing angle, solar zenith angle, relative azimuth). The 2B-CLDCLASS data is provided temporally, with each timestamp associated to a vertical cloud profile, a latitude/longitude point on the Earth, and some metadata, such as quality flags. Although we could quantize the 2B-CLDCLASS locations to the POLDER grid, doing so would cause substantial quantization error. Instead, we compute ground distances to find nearest neighbors in the POLDER data for each 2B-CLDCLASS observation. As it would be computationally prohibitive to compute these distances between all point pairs, we first use a KD-Tree (Bentley, 1975) to find the top 20 POLDER matches in latitude / longitude space for each CloudSat observation. Although the latitude / longitude grid is far from equal-area, at sufficiently small distances, the effect of Earth's curvature on the distance-ranking approaches zero. We then compute approximate ground distances from the CloudSat observation to these 20 points using the WGS-84 ellipsoid model of the Earth. From these 20 points, we select the closest ones that are found to the northeast, southeast, southwest, and northwest of the Cloudsat observation. These four points are used as interpolation corners, with weights defined by their ground distances. Standard bilinear interpolation leverages the separability of the x- and y-components of the interpolation equation, but these corners are neither on a flat plane nor are they guaranteed to be rectangular. For these reasons, the normalized inverse distance is a more appropriate choice for the corner weighting function.

For a pair of half-orbit files, the above defines a mapping from CloudSat observations to their corners in the POLDER grid. In order to sample patches, we compute all of the latitude intervals which would yield a 100 pixel north-south distance in the POLDER grid. We also compute the latitude-dependent longitude intervals which would yield a 100 pixel east-west distance in the POLDER grid. We uniformly sample latitude intervals, and use our longitude intervals to compute an index from a 100x100 patch into the POLDER grid. As an additional step, we uniformly shift patches east or west by one quarter of the patch window so that labels aren't always found in the center of the patch. All patches are north-aligned; we do not perform any rotation. Only Cloudsat observations which have 4 corners within the patch are kept. If there are fewer than 100 Cloudsat observations in a patch, that patch is discarded. Additionally, occasional quantization or out-of-bounds errors can cause the patch to contain fewer than 100x100 pixels or to contain 101x100 pixels. In these cases, we discard the patch. This process loops until sufficient patches are found for a file. Sufficient valid patches were found for every file.

We validate our alignment accuracy by computing (approximate) ground distance between the latitude and longitude values of the projected Cloudsat observations and the weighted interpolation of the latitude and longitude values of the corresponding corners in the POLDER grid. The mean projection error across both the training and validation datasets is 881 m, with a

maximum error of 2,216 m. This is significantly less than a single pixel in the POLDER grid, which are approximately $6 \times 7$ km$^2$. Sub-pixel alignment accuracy ensures that each pixel in the POLDER grid contains sensor values from the correct location.

## 2.5 Dataset Statistics

ATCS is a large-scale dataset, enabling the training of deep neural networks. The 2B-CLDCLASS vertical cloud profiles are used as labels for the PARASOL imagery, which is used as input to the neural network. The ATCS dataset contains 20,352 labeled instances, and 5,032 instances in the test set, whose labels are withheld. The labeled instances are divided, with an 80/20% training/validation split. On average, in the labeled set, there are 116 labeled locations per 100x100 patch, with a standard deviation of 18, a minimum of 55, and a maximum of 171.

Data were sampled uniformly in the geographic sense, except for latitudes greater than $\pm 80°$, which cause complications with geometric processing. There is great value in this global coverage, as cloud dynamics and appearance vary greatly by region. The difficulty of the cloud segmentation task is also strongly regionally dependent. For example, near the poles, there are high-albedo icy and snowy surfaces, which are more visually cloud-like than typical terrestrial surfaces, complicating the segmentation task (Stillinger et al., 2019).

Cloudiness also correlates strongly with altitude. There is a strong 'class imbalance' between the clouds at various altitudes – a randomly sampled pixel is much more likely to contain clouds at a lower altitude than to contain clouds at the upper altitude range, i.e. near 14 km.

## 3 Approach

Our approach is to learn a functional mapping from multi-angle imagery to per-pixel vertical profiles of cloud occurrence using a deep convolutional neural network (CNN). This had two primary motivations. First, by training a model on labeled nadir pixels and applying it to unlabeled off-nadir pixels, we produce a wide-swath vertical cloud product, which has independent value. Second, the supervised learning setup allows for ablation experiments; various sensor properties (angle, spectral band, polarization) can be omitted from the input data. The resulting change in model skill provides insights into the utility of these properties for estimation of cloud structure. The use of a neural network allows the characterization of highly non-linear relationships between these input fields and the cloud profile outputs, and deep learning outperforms shallow learning, as is shown in Table 2.

We used convolutional neural networks to incorporate spatial context, necessary in order to account for the parallax present in multi-angle data. Convolutions are useful for data which exhibit some form of translational invariance, meaning that some elements of appearance do not change due to translations in image space. Interestingly, convolutions are particularly suitable for satellite data, since it exhibits lower translational variance than ground-level imagery, due to less perspective variation in depth. The spatial context provided by convolutional architectures can capture the apparent shift in location due to parallax present in a cloudy scene, which is useful for the network to predict depth and vertical structure.

## 3.1 Representation

Instances are defined as input / output pairs, where input contains a 100x100 patch of multi-angle, polarimetric imagery from POLDER, as well as the pre-computed weights used for interpolation, and output contains the vertical cloud profiles from 2B-CLDCLASS. Each pixel in the input has data for up to 16 viewing angles. When data is missing for one or more viewing angles, a standard missing value of -1 is used. The median number of available viewing angles over the training and validation sets is 14, and 98% of pixels have at least 13 available viewing angles. At each of the viewing angles, there are nine spectral

bands, three of which measure linear polarization (Table 1). Linear polarization is represented as the $Q$ and $U$ components of the normalized Stokes parameters (also known as $S_2, S_3$). Therefore, the channels without polarization have one value: $I$, while the polarization channels have three: $I, Q$, and $U$.

Raw sensor inputs must be converted into features which can be used by a neural network. Most inputs can simply be normalized before use, but some inputs must be more heavily modified. In addition to the nine intensity values $I$, and the three

pairs of $Q, U$ values, there are four geometry fields: viewing angle, solar zenith angle, relative azimuth, and solar azimuth. We found there to be little difference from using view azimuth instead of relative azimuth. Rotations are discontinuous at $360°$, which poses issues for a neural network as a small change in rotation can cause a large change in the output. We use a technique from the object recognition literature which encodes angles into binary membership in two of eight overlapping bins, as well as two in-bin floating point offsets (Mousavian et al., 2017), resulting in a length-ten vector. The eight overlapping intervals

are $[0°, 90°], [45°, 135°], \ldots, [315°, 45°]$. The in-bin regression values are relative to the bin centers. As an example, consider the angle $100°$. This occupies the second bin $[45°, 135°]$ and the third bin $[90°, 180°]$, giving it a bin-membership encoding of 01100000. Its differences with the centers of the second and third bins are $10°$ and $-35°$, respectively. This yields a length-ten feature vector: $\{0, 1, 1, 0, 0, 0, 0, 0, 10°, -35°\}$, although angles are converted to radians in practice. This transformation is only necessary for the azimuth angles, not the zenith angles, as zenith angles are bounded in the range $[0°, 90°]$.

There are 27 input values per view: six non-polarized channels, three polarized channels with three values each (I,Q,U), a length-10 view azimuth feature, one view zenith angle, and one solar zenith angle. The length-10 solar azimuth feature does not vary with view angle. Therefore, when using all 16 available angles, the channel depth (features per pixel) is 442. In addition, there are (non-angular) fields included as a convenience for other researchers. These values are not provided as an input to any model trained in this study, and include latitude, longitude, surface altitude, and a land-or-sea flag.

Each input patch is represented as an image cube, created by stacking the multi-spectral, multi-angular features, as well as the geometry features described above. In line with standard practice in computer vision, each of the spectral features are normalized by subtracting their mean and dividing by their standard deviation, with respect to the training set.

## 3.2 Architectures

Ideally, the selection of model architecture could be derived directly from the scene geometry. The receptive field of a pixel in

the output of a convolutional neural network (CNN) is the extent of the input (in pixels) which can theoretically have affected it. In almost all sufficiently deep CNNs, the receptive field far exceeds the input image size. One could establish a relationship

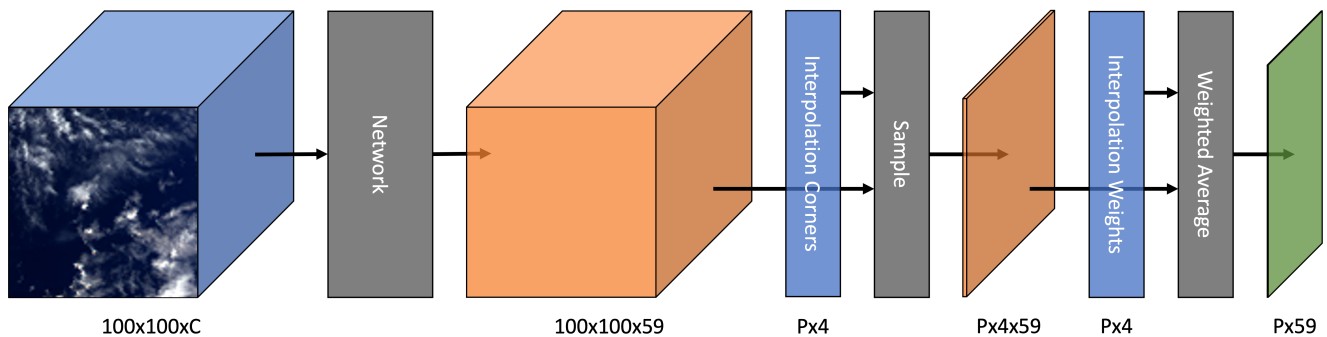

**Figure 2.** Diagram illustrating the forward pass. The color blue denotes inputs, orange denotes intermediate tensors, and green denotes output. C denotes channel depth and P denotes the number of labeled pixels in an instance. The model maps the input POLDER-3 imagery to predictions for each of the 59 height bins. These predictions are then interpolated, represented by the Sample and Weighted Average blocks. Simply skipping the interpolation step allows the generation of wide-swath predictions.

between the maximum possible disparity in pixel space due to parallax and derive a network architecture from that. There are two limitations with this approach. First, is that the model may benefit from having broader context about the scene. Second, is that the 'effective receptive field' is always much smaller than the actual receptive field, and is not straightforward to

compute (Luo et al., 2017). For both these reasons, we instead elect to present a few reasonable architectures. We experimented with many more architectures than are presented here, but found these three to be the best representatives of three different reasonable hypotheses.

### 3.2.1    Single-Pixel Network

A simple baseline approach to cloud segmentation is to independently estimate the vertical profile of cloud occurrence at each

pixel. The hypothesis behind this architeture is that spatial context does not matter for this task. A multi-layer perceptron (i.e. a simple neural network) predicts this vertical profile of a single pixel at a time from the sensor observations of that pixel. The single pixel model is implemented using 2D convolutions with kernel size $1 \times 1$. Since the kernel size is always $1 \times 1$, the independence of pixels is preserved (i.e. the network has no spatial context), but has the important property of keeping the same number of pixels per batch as the later experiments using 2D convolutions. Batch size would otherwise be a potentially

confounding variable in any comparison between the single-pixel model and models which ingest an entire image at a time. We test a single-pixel model with three linear layers, with all but the last layer followed by batch normalization (features averaged across each batch) and a rectified linear unit (ReLU), which is defined as $f(x) = x$ for $x \geq 0$, $f(x) = 0$ for $x < 0$.

### 3.2.2    Simple CNN

We also test a simple multi-layer convolutional neural network, with the hypothesis behind this architecture being that spatial

context does matter, but a high depth (number of layers) does not. This model has five $3 \times 3$ convolutional layers. As with

the single-pixel model, all but the last convolutional layer are followed by batch normalization and ReLU. Each convolutional layer includes one pixel of padding, so the image resolution stays constant during the forward pass.

### 3.2.3 U-Net

Finally, we implement U-Net (Ronneberger et al., 2015). The hypothesis behind this model is that both spatial context and depth are important for this task. A U-Net consists of a fixed number of down-sampling 'blocks', followed by the same number of up-sampling blocks, as well as skip connections between blocks at the same level of the spatial pyramid. The down-sampling blocks capture the typical structure of a convolutional neural network for image classification, where each subsequent layer represents a gradual trade-off of spatial resolution for increasing feature depth. U-Net also adds up-sampling blocks, which effectively do the opposite: decreasing feature depth while increasing spatial resolution. This architecture is related to the commonly used encoder-decoder network. Unlike encoder-decoder networks, U-Net has skip connections, which allow the preservation of spatially located features in the up-sampling path. We use five blocks in our U-Net, with the per-block feature depths using the same scheme as described in Equation 1. It is worth noting that increasing the channel depth in this way substantially affects both U-Net's parameter density and the size of the intermediate features during the forward pass. We experimented with several different schemes to decide channel depth, and found this approach to be the most stable configuration for U-Net.

### 3.2.4 Channel Depths

The number of input channels to our network is approximately two orders of magnitude higher than ground-level imagery: RGB imagery has three channels, but our multi-angle, multi-spectral images (which include a geometry encoding) have hundreds of channels. For example, the 8-angle multispectral experiments have 226 input channels.

A common choice for choosing channel depths in CNNs is to use increasing powers of two, for historical reasons related to the use of pooling operators. This would result in unreasonably high feature depth, even after a few convolutional layers. Instead, we compute a scaling factor which yields a desired depth after a certain amount of layers. These layer-wise feature depths $c_i$ are given by the following, where $c_{\text{input}}$ is the input channel depth, $c_{\text{output}}$ is the desired output channel depth, and $\lfloor \cdot \rfloor$ is the floor operator:

$$c_i = \lfloor c_{\text{input}} \wedge (1 + (\frac{\log(c_{\text{output}})}{\log(c_{\text{input}})} - 1)\frac{i}{5}) + 0.5 \rfloor \tag{1}$$

For the simple convolutional network, $c_{\text{output}}$ is the number of height bins: 59, and the input depth is used for the first layer's feature depth. Thus for our 5-layer CNN, the per-layer depths would be 226, 173, 132, 101, and 59. For the U-Net, $c_{\text{output}}$ is the channel depth after the decoder half of the U-Net, which we set to be 1024. Again, for an input depth of 226, the feature depths would be 306, 414, 560, 757, and 1024. This scaling ensures a consistent rate of change in the feature depth, rounded to the nearest integer. Note that the base of the log does not matter as long as it is consistent.

### 3.3 Interpolation

One novel element of our network architecture is the use of interpolation during the forward pass of the model. The CloudSat labels are only available for some locations in the input images, and these locations are not quantized to the POLDER grid. Therefore, after the up-sampling blocks, the model has an interpolation layer, using the corners and corner weights described in 2.4. As a weighted average is differentiable, back-propagation can still be used to train the network. After the interpolation layer, the network has two fully-connected layers. Figure 2 illustrates the forward pass, including a depiction of the interpolation process. Note the difference between this and bilinear interpolation: bilinear interpolation is linearly separable. For accuracy, the interpolation model utilized here uses a latitude/longitude grid rather than a Cartesian grid, so the standard separability does not apply.

The networks can be applied to data *without* interpolating. This is as simple as omitting the interpolation module. In Figure 2, the $100 \times 100 \times 59$ tensor can simply be used as the wide-swath prediction. The reduction down to a $P \times 59$ tensor is only necessary for the application of the loss function. Interpolation is slightly more complicated for U-Net, which is discussed in Appendix A.

### 3.4 Training Procedure

In all experiments, the model is trained for 30 epochs using the Adam optimizer (Kingma and Ba, 2017) and the binary cross-entropy loss. The cross-entropy loss is theoretically well-founded; it measures how many bits would be needed to encode an event from the actual probability distribution, assuming the optimal encoding from the learned distribution. Cross-entropy also tends to converge quickly. Let $\sigma(\cdot)$ be the sigmoid operator, $y$ the binary labels, and $x$ the network's unbounded outputs (also referred to as logits). The binary cross-entropy loss is given by:

$$\ell_n = y_n \cdot \log\left(\sigma(x_n)\right) + (1 - y_n) \cdot \log\left(1 - \sigma(x_n)\right) \tag{2}$$

We experimented with data augmentation, including random flips and rotations, but found these to significantly worsen performance, even when accounting for these changes in the geometry features. The purpose of these augmentations is to introduce symmetric and rotational invariance into the model without the need for more data. However, symmetric and rotational invariance are less important from an orbital view, and the model may benefit from memorizing the north-locked perspective inherent to the data. Therefore, we omitted data augmentation from our final experiments.

Each experiment is repeated three times, for reasons discussed further in 4.2. Model checkpoints are saved every 5 epochs. Our implementation uses the Pytorch framework (Paszke et al., 2019). Training is performed using a single NVIDIA Tesla V100 GPU with 32GB of memory. Training plus validation typically takes between two and six hours per experiment.

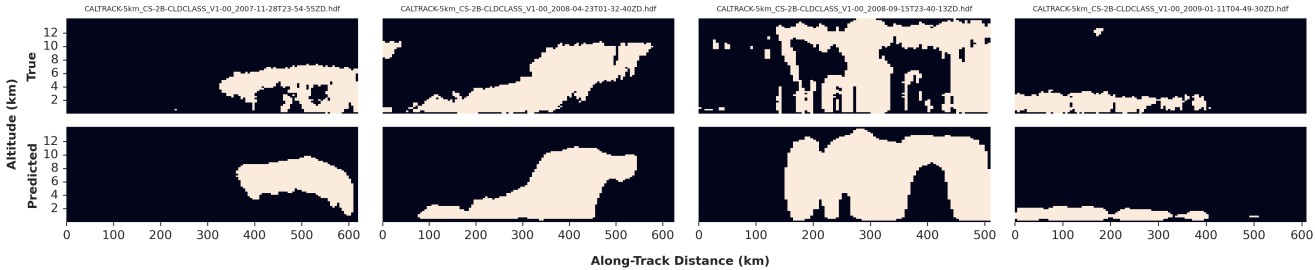

**Figure 3.** Results for four instances in the test set, using the default experiment. The true cloud profiles are shown in the top row, and predictions are in the bottom row. The Caltrack 2B-CLDCLASS file names are included at the top.

## 4 Results

320 We evaluate the model's performance on the test set. Some qualitative results can be seen in Figure 3. More qualitative results are presented in Appendix B. Qualitative results suggests the model is skillful at capturing the larger-scale structure of clouds, with worse performance for smaller cloud segments and multilayer clouds.

We perform a series of ablation experiments. The first experiment compares the skill of the three aforementioned architectures on the test data. In subsequent experiments, we deprive the model of various aspects of the available input data and 325 measure the resulting effect on skill. These ablation studies reveal the impact of the viewing angles, the spectral configuration, polarimetry, and the scene geometry. The baseline 'default' experiment consists of the simple convolutional model with 8 of 16 viewing angles, all available spectral channels (with polarimetry when applicable), and all geometry fields. Other experiments vary from the default experiment only in the exact ways specified in tables 2, 3, C1, and figures 4, 5, and 6.

As the network is trained using the cross-entropy loss, its unbounded outputs (also known as logits) cannot directly be treated 330 as probabilities. Within the loss function, the logits are passed into a sigmoid function to bound them within the $[0, 1]$ range. A threshold of 0.5 after the sigmoid function corresponds to a threshold of 0 before. Therefore, to evaluate the output of the network, we simply threshold the logits at 0 to convert them to a binary value, which we compare with the ground-truth cloud profiles.

### 4.1 Dice Score

335 To evaluate our models, in addition to accuracy, we use the Dice Score. The Dice score, originally used in an ecological context (Dice, 1945), was adopted in the medical segmentation literature as a validation metric for segmentation of MRI imagery (Zijdenbos et al., 1994). Its use has since become standard, due to its several advantages over pixel-wise accuracy metrics, including its tolerance to infrequent positive samples and that it penalizes differences in location more than differences in size. The Dice score is thus a better measure of perceptual quality than pixel-wise metrics. Let $A$ and $B$ be two sets representing all 340 of the discretized locations of two respective objects. The Dice score between $A$ and $B$ is twice the intersection divided by the

sums of the sizes of $A$ and $B$. It can be contextualized in terms of true positives ($TP$), false positives ($FP$), and false negatives ($FN$):

$$\text{Dice}(A, B) = \frac{2 \times |A \cap B|}{|A| + |B|} = \frac{2 \times TP}{2 \times TP + FP + FN} \tag{3}$$

We report the Dice score as a percentage. The Dice score is related to intersection-over-union, also known as the Jaccard index, another metric commonly used in the segmentation literature. Both metrics have a range of $[0, 1]$, but the Dice score is strictly greater than the Jaccard index.

Notably, there is significant evidence that for many applications, the loss function, which is used to optimize the network, should be metric-specific. In segmentation, Jaccard-like loss functions often outperform their pixel-wise counterparts (such as cross-entropy) (Eelbode et al., 2020; Mohajerani and Saeedi, 2021; Wang and Blaschko, 2023). We experimented with Jaccard-like losses early in project development, but observed no apparent improvement over the standard cross-entropy loss.

The Dice score generalizes to an arbitrary number of dimensions. While it is typically used for 2D data, we make use of it in both 1D and 2D contexts. The overall Dice scores presented in the tables are 2D, while the altitude-dependent Dice scores in figures 4, 5, 6, 8, and 11 are 1D, as they describe a single row of the time, altitude cross-section given in the 2B-CLDCLASS labels.

Alongside the Dice score, we report the bin-wise accuracy, which is the rate at which the model correctly assigns a pixel, altitude-bin pair as cloudy or not cloudy. This metric is less strict than the Dice score, and is less suited to labels with a strong imbalance between positives and negatives, as in our data. Our findings are therefore mostly based on the Dice score, but the inclusion of accuracy allows for easier comparison with other works.

$$\text{Acc}(A, B) = \frac{|A \cap B| + |\neg A \cap \neg B|}{|A| + |\neg A|} = \frac{TP + TN}{TP + TN + FP + FN} \tag{4}$$

### 4.2 Inter-run Variability

Due to the stochastic nature of machine learning, the same experiment will yield variable results given different initializations of the network, as well as differences in the shuffled order of the training set. In this work, the inter-run variability is an important factor to consider. Repeating the U-Net experiment 3 times, for example, yielded validation set Dice scores of 72.8%, 73.1%, and 73.4%, although the simpler architectures experience less variance. We report the maximum test set accuracy over three runs; for all runs of an experiment, for all saved model checkpoints, we use the model which yields the highest Dice score on the validation set, and report its Dice score on the test set. There is little variation between the validation set and the test set metrics: U-Net, for example, has a max validation set Dice score of 73.4%, which drops to 73.2% on the test set.

### 4.3 Architecture Complexity

The three architectures described in 3.2 are evaluated, with results shown in Table 2. There is a large increase in skill from the single pixel model to the simple convolutional model, and a negligible increase from the simple convolutional model to the U-Net.

**Table 2.** Comparison of three architectures of increasing complexity. Results are reported for the test set. U-Net achieves the highest performance but has more parameters.

| Architecture | # Params. | Dice Score (%) | Accuracy (%) |
|---|---|---|---|
| Single-Pixel | 1.2E+05 | 68.6 | 93.7 |
| Simple ConvNet | 1.2E+06 | 73.0 | **94.3** |
| U-Net | 9.5E+07 | **73.2** | **94.3** |

**Table 3.** Comparison of various viewing geometries, using the simple convolutional model. The 2-angle view contains only the pair of angles closest to nadir. The 4-angle view adds the next-innermost pair of angles, proceeding outward until all 16 angles are included.

| # View Angles | Dice Score (%) | Accuracy (%) |
|---|---|---|
| 2 | 68.9 | 93.6 |
| 4 | 71.6 | 94.1 |
| 6 | 72.2 | 94.2 |
| 8 | 73.0 | 94.3 |
| 10 | 73.2 | 94.3 |
| 12 | 73.8 | **94.5** |
| 14 | **74.1** | **94.5** |
| 16 | 73.8 | 94.4 |

As the POLDER-3 data are geo-referenced using the surface elevation (and not the cloud-top height), there is a parallax-induced shift when clouds are present, particularly if those clouds occur at a higher altitude. The single-pixel model, which lacks spatial context, does not have access to adjacent pixels. The simple convolutional network and the U-Net, by contrast, can leverage surrounding pixels, making use of the information contained in the parallax. Another potentially useful quality of spatial context is that it captures more information about the scene dynamics, which may be used by the model.

The results suggest that there are diminishing returns from increasing model capacity. It is unlikely, in our estimation, that larger models will achieve significantly higher performance on this dataset, without changes in other aspects of the data processing pipeline or optimization procedure.

## 4.4 Viewing Angles

Multi-angle POLDER measurements are particularly sensitive to parallax, providing useful information on vertical cloud distribution. The impact of parallax on model skill was studied via an ablation experiment in which various viewing geometries were provided for the model. First, we started with an experiment using only the 2 central (closest to nadir) viewing angles.

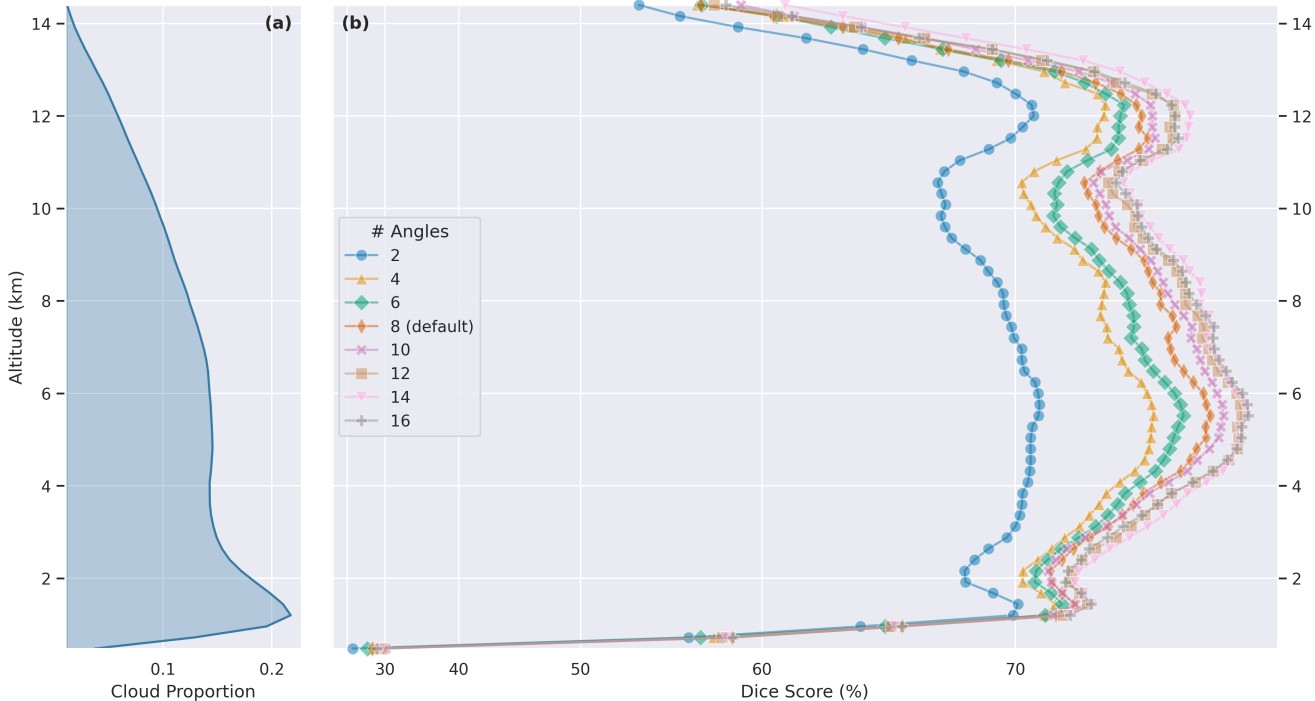

**Figure 4.** Experiments using the simple convolutional model with different viewing geometries. Panel (a) shows the altitude-dependent proportion of pixels labeled as clouds in the dataset. Panel (b) shows the altitude-dependent Dice Score of experiments with different viewing geometries. Note the significant difference between the 2- and 4-angle experiments, with diminishing returns with more angles.

Next, we progressively added subsequent pairs of angles (in increasing zenith angle order) until all angles were included, running independent experiments on each configuration. The results are shown in Table 3, as well as Figure 4, which captures the altitude-dependent Dice score of experiments using various numbers of angles. There is a significant improvement between 2 and 4 angles, with diminishing returns from the inclusion of more angles. There is a clear increasing trend, with 14 viewing angles achieving the best performance. The drop in performance from including the outermost pair of angles is unsurprising, as these angles contain little to no valid data in most scenes, due to the viewing geometry of POLDER-3. We elected to use 8 angles as the default for other experiments, as it marked a good trade-off between performance and training time.

An important factor to consider in the evaluation of these results is the difference between the 2B-CLDCLASS vertical resolution of 240 m with the $6 \times 7$ km$^2$ horizontal resolution of POLDER-3. Even an extreme difference in viewing angles may not be enough to overcome the stark resolution difference. The multiple angles may aid the model to constrain the possible reflectance distribution functions attributable to a surface, helping to identify surface type and altitude, rather than being useful in the stereoscopic sense. This offers one potential explanation for the diminishing return from including more angles, but other explanations remain possible.

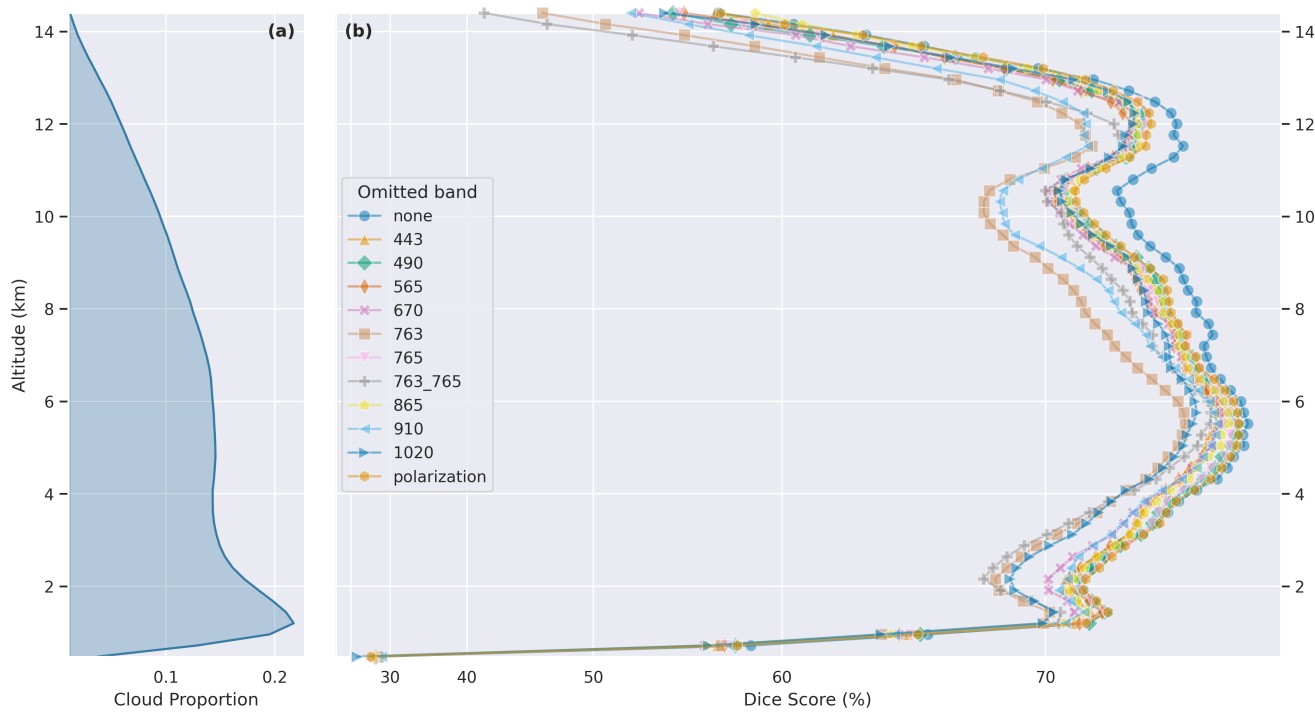

**Figure 5.** Panel (a) as in Figure 4. Panel (b) shows the altitude-dependent Dice Score of experiments with various channels omitted, as compared to the default, labeled 'none'.

## 4.5 Spectra

We perform two ablation studies to understand the impact the nine spectral bands have on model skill. In the first, shown in Figure 5, one or two bands are omitted at a time, and the resulting decrease in model skill provides a measure of the unique (non-redundant) information content present in that band. In the second, shown in Figure 6, only one or two bands are provided at a time, and the model skill represents the bulk information content of that single band. In both studies, we include the 763 nm (oxygen A-band) and 765 nm bands, as well as a combination of both 763 nm and 765 nm band, as they are often jointly used to derive the oxygen pressure within an atmospheric column, which can be used to infer cloud structure (Ferlay et al., 2010). A table of all the spectral experiments can be found in Appendix C, Table C1.

Figure 5 demonstrates the utility of various bands, with larger drops in performance suggesting that the omitted band contains uniquely useful information content. The oxygen absorption feature at 763 nm is highly correlated with model skill, consistent with the known relationship between this feature and cloud structure. The next greatest drop in skill is captured in the 910 nm and 1020 nm band omission experiments. 910 nm is a water vapor band, which provides information on scattering and absorption interactions between clouds and vapor (Dubuisson et al., 2004). The near-infrared 1020 nm band might be useful for low-altitude clouds over the ocean, as clouds are quite bright in this wavelength, while oceans are quite dark.

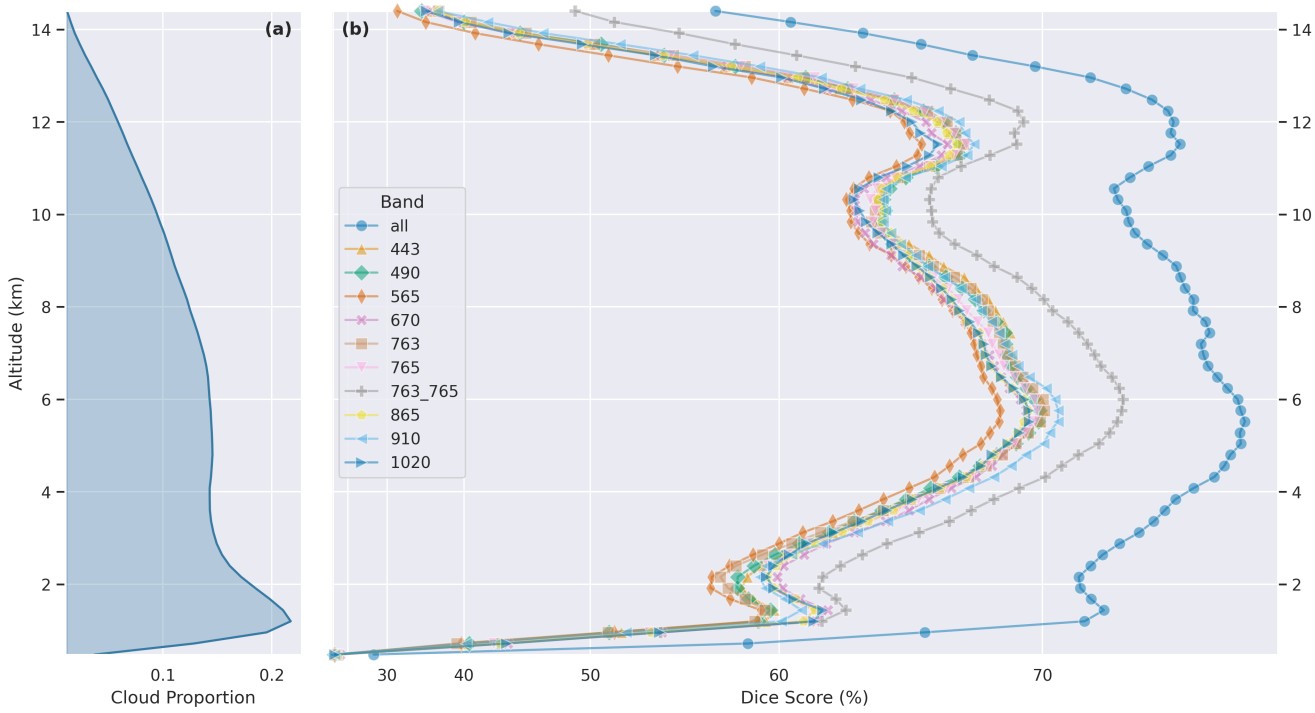

**Figure 6.** Panel (a) as in Figure 4. Panel (b) shows the altitude-dependent Dice Score of experiments using only one or two channels at a time, as compared to the default, labeled 'all'.

## 4.6 Polarimetry

The importance of polarization to model skill is evaluated by comparing the default experiment to one without the polarization parameters. This is achieved by omitting the $Q$ and $U$ channels from the $I,Q,U$ Stokes parameterization. This caused a reduction in Dice Score from 73.0% to 72.6%. This difference is smaller than we hypothesized. One possibility is that the Stokes vector is not the ideal parametrization for this task. Another possibility is that the particular polarization channels in POLDER-3 have only limited unique utility for the derivation of cloud structure. Prior work has found that blue-light polarization is disproportionately useful for the retrieval of aerosol layer height, especially with wavelengths of 410 nm or lower (Wu et al., 2016), but POLDER-3's shortest-wavelength polarization band is 490 nm. The shorter-wavelength polarization information from future missions, like the Hyper-Angular Rainbow Polarimeter (HARP2) aboard PACE (Werdell et al., 2019), might prove more useful. For example, it has been shown that the polarized 440 nm band in HARP2 provides some sensitivity to aerosol altitude (Gao et al., 2023).

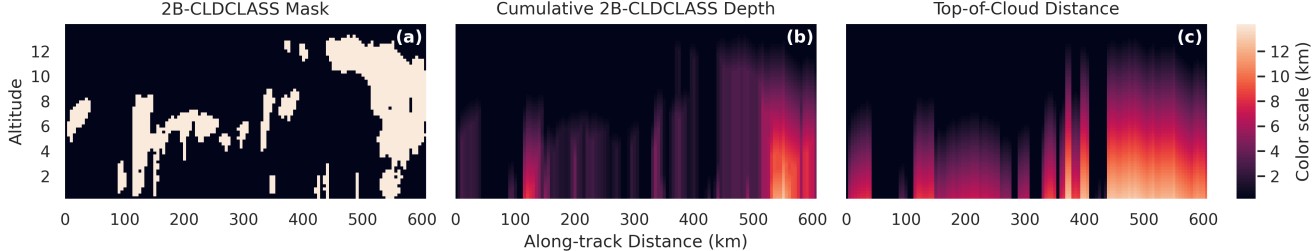

**Figure 7.** Panel (a) as in Figure 3, showing the 2B-CLDCLASS mask for a single instance. Panel (b): cumulative depth for each cell in the mask. Panel (c): top-of-cloud distance for each cell in the mask. Panel (b) and panel (c) differ when there are non-cloudy areas beneath-cloud.

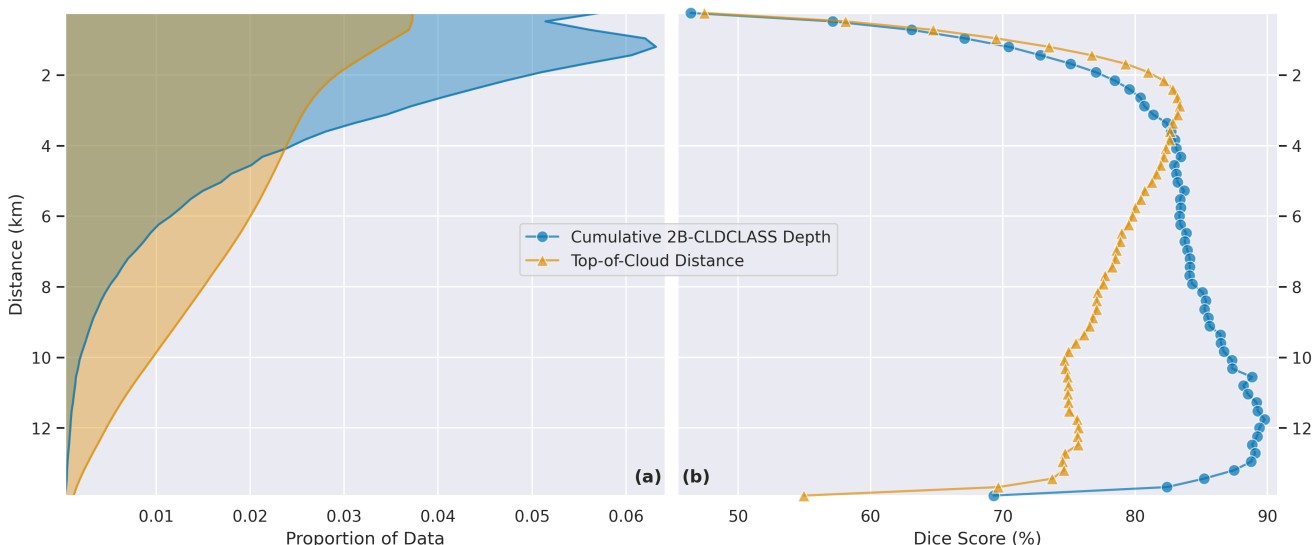

**Figure 8.** Panel (a): Dice score for bins by cumulative 2B-CLDCLASS depth. Panel (b): Dice score for bins by top-of-cloud distance.

## 4.7 Cloud Extent

The vertical and horizontal extent of the cloud being observed is related to the model's skill. Vertical extent is evaluated by stratifying results by two variables: cumulative 2B-CLDCLASS depth and distance to top of cloud. Cumulative depth is simply the cumulative sum of 240 m cloudy bins along each column in the 2B-CLDCLASS product (starting at top of atmosphere), multiplied by the vertical extent of each bin (240 m). Distance to top of cloud is the distance between a predicted bin and the top of the topmost cloudy bin in 2B-CLDCLASS for that column. Figure 7 illustrates the cumulative depth and top-of-cloud distance for a single instance. It is important to note the distinction between cumulative 2B-CLDCLASS depth, optical depth, and geometric depth. First, the 2B-CLDCLASS depth is more akin to geometric depth than optical depth, as the (binarized)

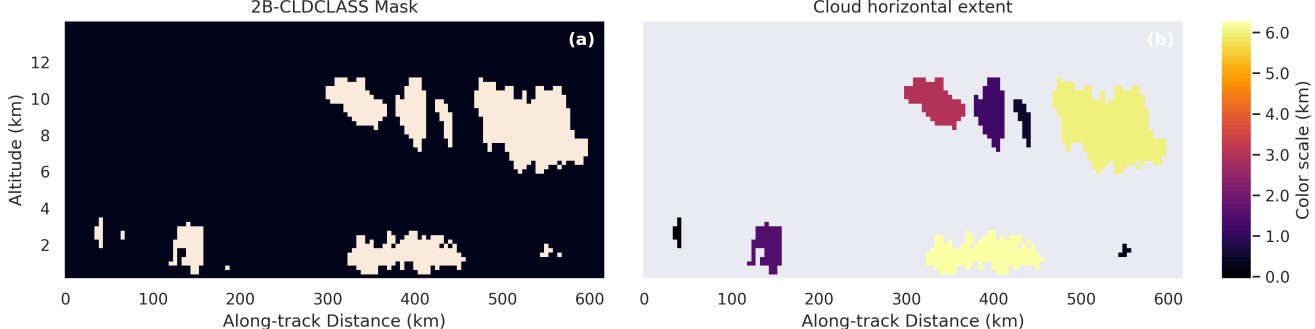

**Figure 9.** Panel (a): cloud mask for an instance in the test set. Panel (b): corresponding cloud horizontal extent of the connected components in the cloud mask. The smallest cloud components are nearly invisible due to their low horizontal extent.

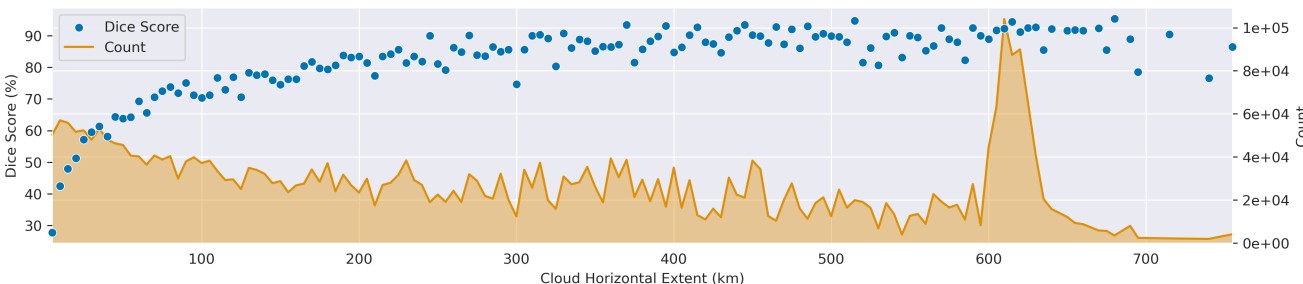

**Figure 10.** Relationship between cloud horizontal extent, counts, and Dice score. The count indicates how many locations in the 2B-CLDCLASS product were assigned to each cloud horizontal extent bin. Dice score indicates the default model's skill over each horizontal extent bin. Note that the spike in count near 615 km corresponds to clouds that exceed the extent of their instances, which are truncated.

2B-CLDCLASS cloud type product does not discriminate between optically thin and optically thick clouds. Still, it is not quite the same as geometric depth, either, as it is discretized to 240 m bins.

Results on the test set are binned according to both the cumulative depth and top-of-cloud distance, and the Dice score is computed for each bin. Figure 8 shows both the proportion of data and the Dice score for bins in both variables. While performance increases with cumulative cloud depth, it decreases with top-of-cloud distance. This indicates that the model is good at identifying very tall cloud systems and struggles with identifying multilayer clouds. These findings are consistent with the qualitative examples in Appendix B.

Quantifying horizontal extent is more complicated. In order to do this, we utilized a connected components algorithm (Bolelli et al., 2020) on the 2B-CLDCLASS mask. A connected component in the cloud mask can be understood as a set of cells where every pair of cells in that set can be mutually reached via only horizontal and vertical steps without crossing a non-cloudy pixel. We then measure the horizontal extent of each component. Figure 9 shows an example of horizontal extent computed on an instance in the test set, while Figure 10 shows the relationship between horizontal extent and model skill. Note that

there are some limitations to this type of analysis. First, this only considers the along-track horizontal extent of the cloud, and assumes that clouds are approximately (horizontally) circular. Clouds which are short in the along-track direction and long in the cross-track direction, or the opposite, will skew these results. A second limitation of this analysis is that clouds whose horizontal extent exceeds the extent of their corresponding instance in the ATCS dataset will be truncated. Note the sharp spike in the counts in figure 8, which corresponds to the most common along-track extent of instances in our dataset, a result of our 100 by 100 pixel sampling size in POLDER-3 and the geometric relationship between the POLDER-3 and 2B-CLDCLASS products. The distribution of cloud horizontal extent approximates the power-law relationship found in prior work (Wood and Field, 2011).

It is clear from Figure 10 that model skill improves as cloud horizontal extent increases. This trend is especially strong at lower horizontal extents, suggesting that the model's skill is not primarily related to parallax. This may result from the dynamics we use to train the model. Larger clouds will exert more influence on the model's supervisory signal than small clouds, even if those smaller clouds would more easily be distinguished from their background with stereo methods. Performance on only the smallest clouds (horizontal extent < 10 km) may suffer due to the limited resolution of POLDER-3. Increases in the resolution of both the passive and active sensor, such as the AOS mission, should allow a similar approach to attain higher performance on clouds with lower horizontal extent.

Stereo methods are expected to perform better on high clouds (more observable parallax) with lower horizontal extent (distinct features). As these are two weakpoints of our method, an ensemble model could offer a promising avenue of study.

## 4.8 Terrain

Performance varies depending on terrain. Distinguishing clouds from the surface using visible or near-infrared channels is more difficult over brighter terrain and easier over the ocean, which is dark. The POLDER-3 data products contain a flag indicating whether each pixel is 'Land' (100), 'Sea' (0), or 'Mixed' (50). This terrain flag is interpolated from the POLDER-3 grid to the 2B-CLDCLASS grid with the method described in 3.3. Any value greater than 0 and less than 100 is treated as 'Mixed.' Figure 11 shows results stratified by terrain type and altitude. The gap between 'Land' and 'Sea'/'Mixed' performance is greatest at altitudes below 3km, where parallax alone is likely insufficient to distinguish clouds from the surface, due to the limited spatial resolution of POLDER-3 data.

## 5 Discussion

The results demonstrate the feasibility of estimating 3D cloud structure from passive, multi-angle imagery. The inclusion of spatial context significantly improved results, while the improvement from using a more complex model was modest (Table 2). Increasingly extreme view angles exhibit diminishing returns (Fig. 4). It is possible that stacking the view angles in the channel dimension might not be the most effective feature representation for retrieving 3D structure. The spectral results (Figs. 5, 6, Table C1) confirm the known utility of the oxygen-A band, water vapor band, and the near infra-red for the study of clouds.

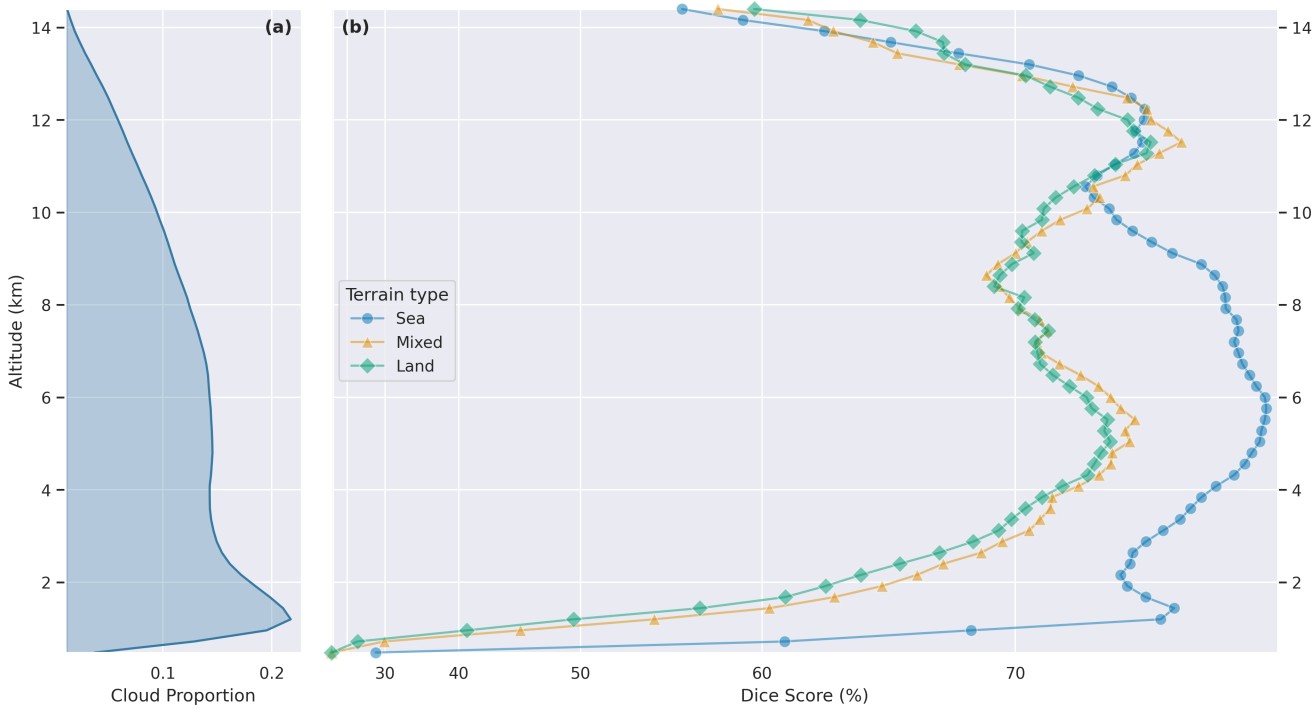

**Figure 11.** Panel (a) as in Figure 4. Panel (b) shows the altitude-dependent Dice score of the default experiment, separated by terrain type.

Polarization only proved modestly useful for the cloud profiling task (Fig. 5, Table C1), but this may be a limitation of the specific polarization bands available in POLDER-3 data.

475     Even the best experimental configuration only reaches a Dice score of 74.1%, which is slightly lower than average results in the segmentation literature (Eelbode et al., 2020). Seemingly, this dataset is challenging, and it is likely that there exists a limit to a model's skill. This should be unsurprising, given the nature of the labeled data: in most segmentation results, the labels come from human annotators. The human annotation process guarantees that the labels are predictable from the imagery, up to the skill of the human annotator. However, our dataset is not annotated by humans, but involves the fusion of two different

480 sources of satellite data, which are sensitive to inherently different physics. This manifests in several ways. For example, one notable characteristic of the qualitative results is that all models struggle to mask the lower layer of multilayer clouds, or to predict the extent of optically thick clouds. The penetration depth of clouds is quite low in the visible and near-visible spectrum. It is possible the models may not have any inputs that allow correct classification beneath cloud tops. Related to this problem is the performance drop across all models at low altitudes. Some of this skill decrease is likely due to penetration depth, but

485 may also relate to lower parallax near the surface, making it harder for the model to vertically resolve cloud location. Another phenomenon this approach may struggle to capture is optically thin clouds. As the labels are based on a cloud fraction product and not an optical depth product, there are extremely optically thin clouds present in the data. The information contained in the visible and near-visible wavelengths may not be enough to detect such clouds.

One of the most important findings of this work is the importance of spatial context. As we move into the next generation of multi-angle sensors with PACE and AOS, we can expect the resolution of these products to improve. In order to achieve the same spatial context in these higher resolution products, the convolutional architectures will require more layers. U-Net is likely not worth the high number of parameters for POLDER-3 data, but it may be a more optimal choice for PACE and AOS. In addition to finer spatial resolution, the HARP2 polarimeter aboard PACE provides better polarimetric accuracy than POLDER-3, but lacks its absorption channels. Evaluating the impact of this tradeoff on 3D cloud masking skill should yield valuable insights. The EarthCARE mission (Wehr et al., 2023) will have periodic simultaneous overpasses with PACE, and its two active instruments will enable the application of similar, partially supervised techniques.

## 6 Conclusions

We designed a supervised machine learning method to perform 3D cloud masking over a wide swath from multi-angle polarimetry, and introduced a dataset to support this method. The dataset should be useful for future research in this area, as it is designed for ease-of-use in machine learning applications. The code accompanying the dataset includes everything necessary to reproduce the results in this paper: to train and validate models, generate figures, and even to create custom datasets.

By performing extensive ablations with various model inputs and hyperparameters, we analyzed the qualities of both POLDER-3 and CloudSat CPR data, as well as their relationship. Our conclusions both confirm existing knowledge and offer new insights. We found a strong relationship between the number of angles and model skill, with the 2-angle and 14-angle variations of our default experiment achieving Dice scores of 68.9% and 74.1% respectively, a 5.2% difference - confirming the value of multi-angle sensors. The multispectral nature of the POLDER data is also strongly related to skill, with the oxygen absorption band proving particularly important. Omission of the oxygen absorption bands alone resulted in a drop in the Dice score from 73.0% to 71.0%, and there is a 6.1% difference in Dice score between the best performing single-band experiment (66.9%) and the default multi-band experiment (73.0%). Our method works well for optically thick and horizontally long clouds, unlike stereo methods, suggesting the two approaches may have some synergy. In particular, model skill was worse for clouds below $10 \, \mathrm{km}$ altitude over land surfaces, while it was similar for land and sea surfaces above $10 \, \mathrm{km}$. Finally, model skill was predictably worse for multilayer clouds, but surprisingly high for very tall cloud systems. Overall, results are promising, and suggest the continued use of machine learning as a means to understand the relationships between various sensor modalities. Additionally, machine learning might, with further refinement, offer a useful way to retrieve 3D cloud masks from satellite data at an unprecedented scale, which would be an invaluable source of data for climate modeling.

This study constitutes an initial foray into the combined use of machine learning and multi-angle polarimetry for 3D cloud masking. Whereas its use here provided insights on the POLDER-3 sensor, other sensors have yet to be studied in such a way. The recently launched PACE mission and the upcoming AOS mission will carry multi-angle polarimeters, providing a useful testbed for this approach.

The model's supervision in this study was provided by an active radar, constrained to nadir locations. The off-nadir performance of our approach has yet to be validated, and would likely require cross-referencing with ground-based radar. The

validation of its 2D (flattened) accuracy could be performed with other wide-swath cloud products, like the MODIS cloud mask (Ackerman et al., 2015). Having such a wide-swath 3D cloud mask product, were its accuracy sufficient, could prove useful for climate modeling.

Alternative approaches from the computer vision literature may be better-suited to the stereoscopic nature of this data. 3D reconstruction pipelines such as COLMAP (Schönberger and Frahm, 2016; Schönberger et al., 2016) might be adapted for wide-swath multi-angle imagery, as has been done for high-resolution satellite imagery (Zhang et al., 2019). These would allow the estimation of 3D cloud structure without the need for radar-based supervision.

*Code and data availability.*  Both code and data will be made available prior to publication, and can be retrieved using SeaBASS, found at
the following link: https://seabass.gsfc.nasa.gov/search, by searching for "ATCS".

## Appendix A: U-Net Forward Pass

Interpolation during the forward pass is more complicated for the U-Net model, as illustrated in Figure A1. During development, we found that the U-Net model performed better if it included fully connected layers after the interpolation module. However, these fully connected layers can still operate on the full (not interpolated) features, of shape $100 \times 100 \times C$. By
flattening this to a $10000 \times C$ tensor, passing it into the fully connected layers to get a $10000 \times 59$ tensor, and unflattening it to $100 \times 100 \times 59$, the network can be applied to wide-swath data. At training time, the interpolation module is applied to the $100 \times 100 \times C$ tensor to retrieve a $P \times C$ feature, which is then passed through the fully-connected layers to get a $P \times 59$ tensor, which is then compared with the ground-truth labels in the loss function.

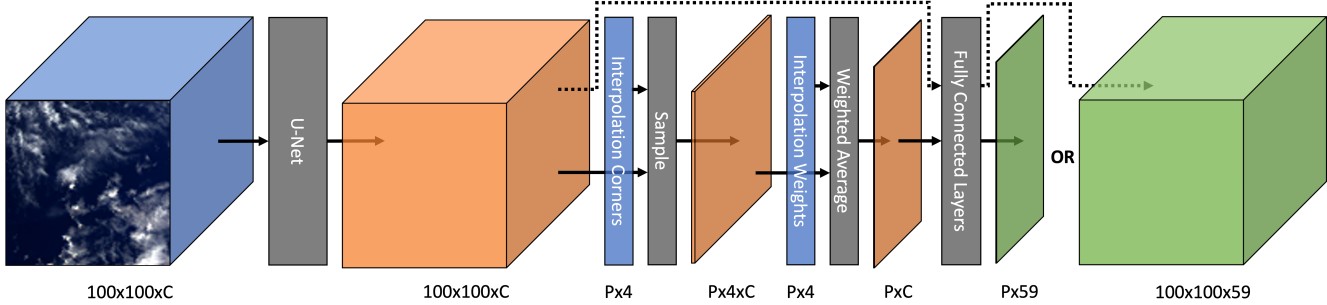

**Figure A1.** Diagram illustrating the forward pass. The color blue denotes inputs (the shown image is from POLDER-3), orange denotes intermediate tensors, and green denotes output. C denotes channel depth and P denotes the number of labeled pixels in an instance. The dotted line represents an alternate pathway which skips the interpolation module, used to get wide-swath results.

# Appendix B: Qualitative Results

We include more qualitative examples in Figure B1, for reference.

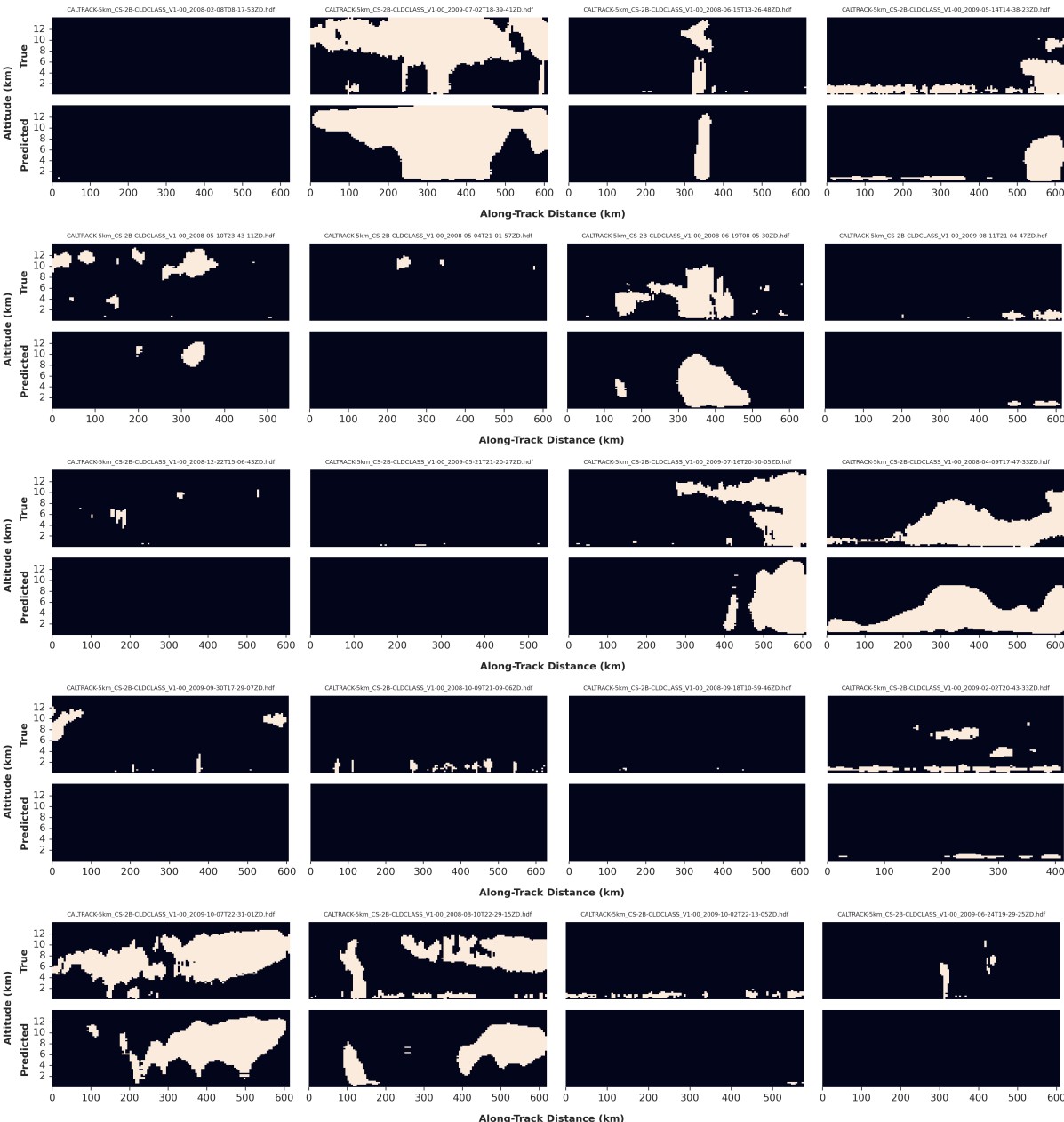

**Figure B1.** Results for twenty instances in the test set, using the default experiment. The true cloud profiles are shown in the top row, and predictions are in the bottom row

# Appendix C: Spectral Results

The full table of spectral results C1 is included here for reference. The experiments here are the same ones illustrated in Figure 5 and Figure 6.

**Table C1.** Results for a variety of experimental configurations. The first section shows with and without polarization, the second section shows the effect of omitting each spectral band, and the third section shows the results when only one band is included at a time.

| 443 | 490* | 565 | 670* | 763 | 765 | 865* | 910 | 1020 | Polarization? | Dice Score (%) | Accuracy (%) |
|:---:|:---:|:---:|:---:|:---:|:---:|:---:|:---:|:---:|:---:|:---:|:---:|
| ✓ | ✓ | ✓ | ✓ | ✓ | ✓ | ✓ | ✓ | ✓ | ✓ | 73.0 | 94.3 |
| ✓ | ✓ | ✓ | ✓ | ✓ | ✓ | ✓ | ✓ | ✓ |  | 72.6 | 94.2 |
|  | ✓ | ✓ | ✓ | ✓ | ✓ | ✓ | ✓ | ✓ |  | 72.1 | 94.2 |
| ✓ |  | ✓ | ✓ | ✓ | ✓ | ✓ | ✓ | ✓ |  | 72.4 | 94.2 |
| ✓ | ✓ |  | ✓ | ✓ | ✓ | ✓ | ✓ | ✓ |  | 72.0 | 94.2 |
| ✓ | ✓ | ✓ |  | ✓ | ✓ | ✓ | ✓ | ✓ |  | 71.8 | 94.2 |
| ✓ | ✓ | ✓ | ✓ |  | ✓ | ✓ | ✓ | ✓ |  | 70.3 | 93.8 |
| ✓ | ✓ | ✓ | ✓ | ✓ |  | ✓ | ✓ | ✓ |  | 72.4 | 94.2 |
| ✓ | ✓ | ✓ | ✓ |  |  | ✓ | ✓ | ✓ |  | 71.0 | 94.0 |
| ✓ | ✓ | ✓ | ✓ | ✓ | ✓ |  | ✓ | ✓ |  | 72.2 | 94.2 |
| ✓ | ✓ | ✓ | ✓ | ✓ | ✓ | ✓ |  | ✓ |  | 71.3 | 94.0 |
| ✓ | ✓ | ✓ | ✓ | ✓ | ✓ | ✓ | ✓ |  |  | 71.3 | 94.0 |
| ✓ |  |  |  |  |  |  |  |  |  | 64.1 | 92.9 |
|  | ✓ |  |  |  |  |  |  |  |  | 64.0 | 92.9 |
|  |  | ✓ |  |  |  |  |  |  |  | 62.9 | 92.7 |
|  |  |  | ✓ |  |  |  |  |  |  | 64.2 | 92.8 |
|  |  |  |  | ✓ |  |  |  |  |  | 63.9 | 92.8 |
|  |  |  |  |  | ✓ |  |  |  |  | 64.3 | 92.9 |
|  |  |  |  | ✓ | ✓ |  |  |  |  | 66.9 | 93.4 |
|  |  |  |  |  |  | ✓ |  |  |  | 64.2 | 92.9 |
|  |  |  |  |  |  |  | ✓ |  |  | 64.6 | 92.8 |
|  |  |  |  |  |  |  |  | ✓ |  | 64.0 | 92.8 |

## Appendix D: Solar Geometry

The relationship between solar geometry and model skill was examined by stratifying the results by the POLDER-3 solar zenith and solar azimuth, with both angular dimensions divided into $5°$ bins. Figure D1 shows these results, including both the test set instance count and the default model's Dice score, for each bin. Most solar geometry bins are empty, due to the sun-synchronous orbit of the A-Train. Local equator crossing time for PARASOL was in the early afternoon during the dataset's time interval (from 2007 to 2009). Solar geometry in this dataset is mostly a function of latitude.

No strong relationship between solar geometry and model skill was found. The only notable drops in Dice score occur in bins with very little corresponding data. This means the results are less statistically significant, but it also suggests that the performance drop results from the model fitting to the dominant mode in the dataset, rather than from the physical properties of observations of the rare geometries.

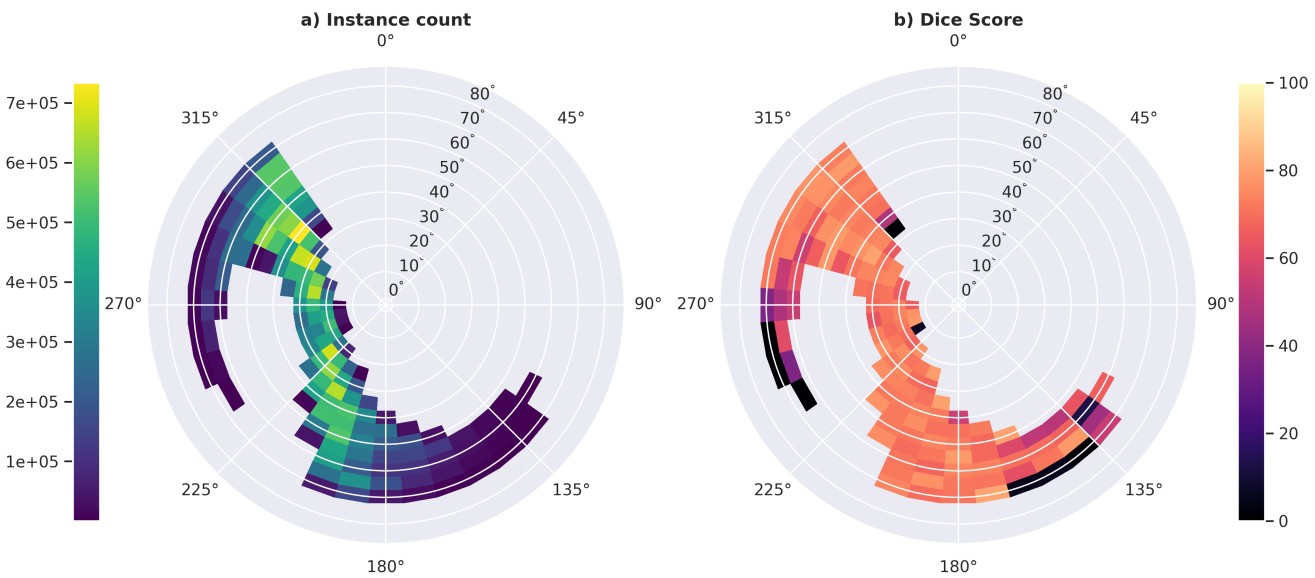

**Figure D1.** Results stratified by solar zenith and azimuth. Panel a) shows the number of instances in the test set in each bin. Panel b) shows the Dice score of the default model's predictions for each bin.

*Author contributions.* SRF developed the code and dataset, and wrote the draft. KDK, AMS, and MG guided the experiments and revised
the paper. JaH and JuH advised the machine learning-related aspect of the project.

*Competing interests.* One of the authors (AMS) is an Associate Editor for AMT. This manuscript was handled by an independent Associate Editor. The other authors declare no competing interests.

*Acknowledgements.* We'd like to acknowledge the team at ICARE for maintaining the POLDER and 2B-CLDCLASS data, as well as the CloudSat team for providing such an invaluable source of vertical cloud data. We'd also like to acknowledge Jie Gong (NASA Goddard Space Flight Center) for useful discussions about the relevant data products. A significant portion of this work was performed during an internship through Science Applications International Corporation (SAIC). That internship was managed by Fred Patt (NASA Goddard Space Flight Center, SAIC), who provided valuable advice.

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
