# Peer review of "3-D Cloud Masking Across a Broad Swath using Multi-angle Polarimetry and Deep Learning"

_EGUsphere, 2023_

## Author Comment (AC1)

Dear reviewers,

We'd like to thank you for the careful consideration of our paper. We have incorporated your revisions and believe the paper is much stronger for it. Many suggestions have led to the inclusion of new figures which elucidate the strengths and weaknesses of the proposed technique.

The major changes are listed below. Some minor changes are also listed below; the rest are listed with the responses to reviewer comments.

**Major Changes**

- Added a new section (4.7) and four new figures on the relationship between cloud horizontal/vertical extent and model skill.
- Added a new section (4.8) and a new figure to illustrate the relationship between terrain/region type (land, sea, mixed), altitude, and model skill.
- Added a new appendix (B) and a new figure (B1) with more qualitative examples.
- Added a new appendix (D) and a new figure (D1) which delves into the relationship between solar geometry and model skill. We found that solar geometry and model skill were largely uncorrelated.
- Moved the part about per-layer feature depths to its own section, correcting a mistake about the simple CNN depths

**Minor Changes:**

- Changed "open-source" to "publicly available" when describing the dataset in the abstract
- Added missing references to tables and figures
- Added a paragraph at the start of section 4 discussing Figure 3.
- Filenames have been added to the qualitative figures (3 and B1)

**RC1**: 'Comment on egusphere-2023-2392', Anonymous Referee #2, 05 Feb 2024
General Comments:

This work trains a hierarchy of deep learning models to predict 3D cloud volume. It uses multi-angle, multi-spectral polarimetric imagery as the input and is trained against W-band cloud-radar measurements. This paper is a good fit for atmospheric measurement techniques, as it demonstrates a new technique for processing multi-angle observations that will benefit several upcoming satellite missions. The methodology also appears sound. However, I think that the results can be explored in more depth to provide more insight into the performance of the technique and the information content of the measurements. I therefore recommend major revisions to the paper to accommodate additional analysis suggested below.

Specific Comments:

The performance of the technique (dice score) is only categorized by altitude which provides limited insight into the factors controlling the technique and the information content of the measurements. I suggest additional analysis below to solidify the support for several key discussion points in the paper.

The results should be stratified by region/surface type and perhaps solar zenith angle. The authors mention a strong sensitivity to region/surface in principle but do not quantify it in their own results.

We added a new section (4.8) and a new figure (11) which explores surface type.

As for solar zenith angle, we found no meaningful relationship between it and our model's skill (see below plots). However, we did include a new appendix (D) which stratifies results by solar zenith *and* solar azimuth. These results indicate there is little meaningful relationship between solar geometry and skill.

[Figure]

Furthermore, I think that the ability to detect the cloud base/volume in thick or multi-layered scenes should be explicitly explored by quantifying skill as a function of cumulative optical depth from the TOA measured by CALIOP and/or by categorizing by nth layer as identified by CLDCLASS-LIDAR (or similar) product. The authors mention this in their discussion, but it is not quantified. Quantifying this will highlight whether or not the POLDER measurements are able to extract the signal of distinct cloud layers within the same column from their multi-angle signatures and would be a valuable contribution.

Secondarily, I wonder about the role of cloud-horizontal-size, and how it varies by altitude, in setting the performance of the technique. I imagine that performance degrades for clouds that become unresolved by the multi-angle measurements.

It would be valuable if the authors could quantify this as it might provide some guidance on how the performance of the technique may extrapolate to higher-resolution measurements (e.g. PACE/AOS) as more and more clouds will be 'resolved' by the multi-angle imagery.

We added a new section (4.7) and four new figures (7, 8, 9, 10) on the relationship between vertical / horizontal extent and skill.

Our expanded analysis does not use cloud optical depth but rather the geometric depth of clouds from CLDCLASS. This is because we do not have ready access to a reference optical depth data set – the CALIOP lidar saturates at moderate optical thicknesses, and the CloudSat radar profiles are not directly interpretable as a visible-wavelength optical thickness (nor are they readily available on the Caltrack reference grid). We believe that examining physical cloud depth provides sufficient context for the understanding of the behavior of the NN model.

The authors mention poor performance when CALIOP observations are included in the training data due to the abundance of optically thin clouds (Line 105). This seems like an important point, and so it would be useful to describe exactly the properties of these clouds (i.e. what is the definition of 'optically thin'?). Does this include optically thin low clouds – to which passive sensors are actually more sensitive? Or does it primarily result from very thin (tau < 0.3) cirrus clouds?

We updated line 105 to clarify that the issues with CALIOP primarily arise from cirrus clouds.

I also would suggest the opposite issue in the training of the technique may also exist, which is that POLDER is actually sensitive to many clouds that are not flagged by the radar despite having a substantial optical signal (Chan and Comiso, 2011), which may

result in poor skill. My guess is that including CALIOP in the training data may actually improve skill at low altitudes with appropriate filtering of the training data.

This is certainly possible. We added a note in the text, as well as a reference to the Chan and Comiso paper. Modifying the proposed method to work for CALIOP + CloudSat data would be an interesting future project. As it would likely require significant filtering of the training data, as well as potential modifications to the optimization procedure, we consider it beyond the scope of this work.

The authors describe several different network architectures which take stacked multi-angle imagery registered to the surface. My understanding is that a CNN requires wide enough filters and a deep enough number of layers to encode non-local information. When combining multi-angle features there should be a simple relationship between the disparity in pixel-space and the required depth/width of the network. These arguments would suggest a relatively small amount of non-local information available from POLDER, supported by the relative skill of the models, but not for higher-resolution measurements. I think it would be valuable for the authors to comment on the selection of model architectures for these sorts of problems and describe what they might do for (PACE/AOS).

We addressed the comment about architecture selection by adding a paragraph at the beginning of section 3.2 (Architectures) which further justifies why we approached architecture selection in the way we did, including the concept of a receptive field. We also added one sentence to each architecture paragraph explaining the hypothesis behind its selection, in terms of spatial context and model depth. Finally, we added a short paragraph at the end of the discussion section connecting what higher-resolution products in PACE and AOS will mean for architecture selection.

We would slightly disagree about the 'relatively small amount of non-local information available from POLDER.' Yes, there is only a small gap in skill between the simple convolutional network and the U-Net. But even with only 5 convolutional layers (3x3 kernels), the receptive field of the simple convolutional network works out to 11x11, so 66-77km on the surface (although the effective receptive field will be somewhat smaller). We suspect some of the (significant) difference between the single pixel model and the simple convolutional model is due to the addition of this spatial context. The U-Net, by design, has a receptive field that covers the entire image. It is unclear how much of the (small) skill difference between the simple convolutional network and UNet is due to the receptive field / effective receptive field and how much is due to other properties of the architecture. An interesting follow-up work might involve a thorough comparison of many different architectures at this task, to isolate which

principles of CNN design translate from the ground-level imagery domain, and which principles do not.

Technical Comments:

Line 7: The comment about "limiting assumptions" requires some justification as polarized 3D RT is an asymptotically accurate approximation of EM in the atmosphere. Computationally expensive, for sure.

We removed the comment about "limiting assumptions." You are right that it requires more justification. That justification doesn't belong in an abstract.

Line 9: This pre-processed dataset doesn't seem that generally relevant, given that it is a single dataset, rather than a tool that can operate on and harmonize many different datatypes. Perhaps, this should not be emphasized as strongly in the abstract?

We chose to focus on the dataset as an independent contribution as it enables other researchers to more easily follow up on our work. Datasets are often strongly emphasized in machine learning papers, for this reason, and we want to advertise the availability of data to any readers who might otherwise only read the abstract.

Line 19 & 20: Some more specificity/references here would be good e.g. radiative effects / hydrological cycle etc.

This is a good idea; we added a line and a citation about cloud radiative effects in the introduction.

Line 22: "is of utmost importance" rather than "will be"?

Thank you for the catch; this has been corrected.

Line 44: References for multi-layer errors. (Mitra et al., 2021; Holz et al., 2008)

Line 46: Also worth mentioning CPR's & CALIOP's sensitivity issues/differences for thin, liquid clouds (Christensen et al., 2013; Chan and Comiso, 2011)

Line 42 – 56: There is a wide variety of work that employs statistical techniques to retrieve 3D cloud structure beyond Brüning et al., 2023 ranging from local nearest-neighbor matching from Barker et al., 2011 for 3D reconstruction to generative adversarial networks trained on MODIS-A-train pairs (Leinonen et al., 2019; Barker et al., 2011).

Thank you for the references. These have been added to the text.

Line 48: "The introduction of uncertainty as a part of stereo algorithms must be weighed against the benefit of a wider swath." I don't quite understand this sentence. Perhaps, "the introduction of uncertainty into the retrieved cloud top height as a result of applying stereo." If this is the statement, then I would disagree. All retrievals have uncertainties and stereo tends to be more precise than the mono-angle radiometric alternatives. The tradeoff is also in terms of cost. A multi-angle camera is much cheaper than a lidar.

We removed the misleading sentence – it was originally intended to emphasize the uncertainty of stereo but fails to account for the *relative* uncertainty of stereo vs. other retrievals.

Line 62: It would be helpful to reference some examples of these applications, even if they are for surface remote sensing, for example.

Line 65 – 70: (Castro et al., 2020) might also be referenced here as an example of high-resolution cloud stereo.

Thank you, this reference has been included.

Line 63: Putting the exact POLDER resolution here would be helpful.

The exact POLDER resolution has been added.

Line 78: 3DeepCT does not perform a segmentation task. It is not retrieving a binary mask, it is regressing for the 3D liquid water content. The more sophisticated extension of this work may be of interest in terms of model architectures which explicitly handle the multiple projections of multi-angle imagery (Ronen et al., 2022).

You are correct – and this has been corrected. We also added a reference to the Ronen et al. paper.

Line 95: Should this be "16 viewing angles from which a point on the Earth can be observed"?

We changed "pixel" to "point on the Earth."

Line 242: A reference for the increasing powers of two in number-of-filters per layer would be helpful.

Added a short justification, referencing the relationship between powers of two and the down-sampling caused by 2x2 max pooling operations. A discussion of the many reasons for increasing by powers of two would require its entire own section and is beyond the scope of this paper.

References:

Barker, H. W., Jerg, M. P., Wehr, T., Kato, S., Donovan, D. P., and Hogan, R. J.: A 3D cloud-construction algorithm for the EarthCARE satellite mission, Quarterly Journal of the Royal Meteorological Society, 137, 1042–1058, https://doi.org/10.1002/qj.824, 2011.

Castro, E., Ishida, T., Takahashi, Y., Kubota, H., Perez, G. J., and Marciano, J. S.: Determination of Cloud-top Height through Three-dimensional Cloud Reconstruction using DIWATA-1 Data, Sci Rep, 10, 7570, https://doi.org/10.1038/s41598-020-64274-z, 2020.

Chan, M. A. and Comiso, J. C.: Cloud features detected by MODIS but not by CloudSat and CALIOP, Geophysical Research Letters, 38, https://doi.org/10.1029/2011GL050063, 2011.

Christensen, M. W., Stephens, G. L., and Lebsock, M. D.: Exposing biases in retrieved low cloud properties from CloudSat: A guide for evaluating observations and climate data, Journal of Geophysical Research: Atmospheres, 118, 12,120-12,131, https://doi.org/10.1002/2013JD020224, 2013.

Holz, R. E., Ackerman, S. A., Nagle, F. W., Frey, R., Dutcher, S., Kuehn, R. E., Vaughan, M. A., and Baum, B.: Global Moderate Resolution Imaging Spectroradiometer (MODIS) cloud detection and height evaluation using CALIOP, Journal of Geophysical Research: Atmospheres, 113, https://doi.org/10.1029/2008JD009837, 2008.

Leinonen, J., Guillaume, A., and Yuan, T.: Reconstruction of Cloud Vertical Structure With a Generative Adversarial Network, Geophysical Research Letters, 46, 7035–7044, https://doi.org/10.1029/2019GL082532, 2019.

Mitra, A., Di Girolamo, L., Hong, Y., Zhan, Y., and Mueller, K. J.: Assessment and Error Analysis of Terra-MODIS and MISR Cloud-Top Heights Through Comparison With ISS-CATS Lidar, Journal of Geophysical Research: Atmospheres, 126, e2020JD034281, https://doi.org/10.1029/2020JD034281, 2021.

Ronen, R., Holodovsky, V., and Schechner, Y. Y.: Variable Imaging Projection Cloud Scattering Tomography, IEEE Transactions on Pattern Analysis and Machine Intelligence, 1–12, https://doi.org/10.1109/TPAMI.2022.3195920, 2022.

**Citation**: https://doi.org/10.5194/egusphere-2023-2392-RC1

**RC2**: ['Comment on egusphere-2023-2392'](), Anonymous Referee #1, 06 Feb 2024
This work explores the applicability of deep learning models to predict 3D cloud masking. Using multi-spectral, multi-angle polarimetry, three models are trained, each with increasing complexity, to output a vertical profile (radar-based). The proposed approach is novel and combined with the data set makes for a compelling addition to this line of work. However, the limited analyses do not fully exhibit the performance of the models nor do they help in evaluating the influence of geometry. Additionally, the text often does not connect the physical world to the machine learning techniques and terminology and could use revisions to help the general reader understand why such an approach could be seen as beneficial as opposed to traditional ones. If some of the changes listed below are adopted, I believe the paper would be stronger and advocate the approach better. I recommend accepting the paper upon major revisions.

Specific Comments:

While the background and methodology are explained very well, the current iteration lacks a few key points and analyses. Perhaps the most glaring issue lies with the lack of mention or any explanation for Figure 3 which is arguably the most important figure to capture the performance of the models. Based on this figure alone, one could perform more analyses using histograms and heatmaps (details about further analyses below).

Thank you for pointing this out. We added a reference to figure 3 in the text. We also added an appendix (B) with more qualitative examples, and line 315-316 comments on these results. Four new figures (7, 8, 9, 10) also address this comment, to a certain extent.

The Dice score, while familiar to a machine learning (ML) audience, is not a commonly used term in remote sensing or the geosciences. The physical interpretation of the Dice score is not presented in the text which makes it harder to see the use of such a metric. The authors should add more context around the similarity scores and how the Dice score plays into it along with examples of what a Dice score for a given pair of images would mean in a practical sense. In the same vein, the authors report Dice score only as a percentage, which is slightly misleading, especially since accuracy is also another metric employed here. It should be reported as a fraction/decimal and if needed, explained further by converting to percentages.

Thank you for this comment, and we relate to this sentiment. Unfortunately, the Dice score is not very intuitive, as it lacks a fundamental physical interpretation besides its relationship to the intersection-over-union of two sets. Equation 3 presents the Dice score in these terms, as well as relating it to true/false positives/negatives.

The Dice score is used for the reasons discussed in lines 330-333: "… several advantages over pixel-wise accuracy metrics, including its tolerance to infrequent positive samples and that it penalizes differences in location more than differences in size." As the paper is already quite long, we chose not to add more figures to explain the Dice score. The references included in section 4.1 have lengthy discussions of the Dice score.

Dice score is interchangeably represented as a decimal or as a percentage in the image segmentation literature. We find that the difference is not meaningful, except for that it is easier for the reader to compare the accuracy and Dice score numbers when both are expressed as percentages.

Additionally, some of the ML terminology is either not explained in full or lacks a rooting in the physical world (details in the technical comments). On the other hand, one of the central parts of a paper leveraging deep learning is the architecture and while the authors detail this in the text, figures on the architectures of the 3 models (or at least the simple-CNN and U-Net) would be more beneficial. While Figures 2 and A1 help, they are more schematic flows than architectures. It is recommended that another appendix be added to this paper to delve deeper into the ML terminology as well as more details and figures on the architectures and hyperparameters (although one could see the benefit of including a figure of the architecture within the main text).

Thanks for the feedback – sometimes it is difficult to know to what extent to explain some of these ML concepts. We've added some more explanations as you requested in the technical comments.

As for the architecture diagrams, we decided not to include them for U-Net as we did not want to imply that our modifications to U-Net were anything more than trivial. The U-Net paper cited in the text already has a great diagram, and the only important difference in our implementation is the feature depths, which are described in depth in section 3.2.4. A diagram for our simple CNN is not much more informative than simply knowing the feature depths; a 3x3 convolutional kernel with padding of 1 does nothing to change the feature width / height. Only the depth changes. We include a quick diagram of the simple CNN to the right. The tensor's size during the forward pass is shown in black, with the convolutional parameter sizes in the green boxes, listed in order of input depth, output depth, kernel height, kernel width.

100 x 100 x 226

Conv2D(226, 226, 3, 3)

100 x 100 x 226

Conv2D(226, 173, 3, 3)

100 x 100 x 173

Conv2D(173, 132, 3, 3)

100 x 100 x 132

Conv2D(132, 101, 3, 3)

100 x 100 x 101

Conv2D(101, 59, 3, 3)

100 x 100 x 59

Overall, the paper leans heavily into ML with some impressive results. However, I expected more analyses to be performed. For instance, the authors mention that training is performed on various sensor properties but ==do not show any quantifiable analysis on some important factors==. Results based on ==solar geometries== would a) provide insight into the model's invariance or dependence on them, and b) quantify the effect of the geometrical corrections applied earlier in the data set. Another and more important analysis could be based on the ==types of clouds== (stratified by phase, optical thickness, etc.), particularly relevant since instruments like CALIOP are known to have issues with thin clouds and would provide another perspective into instrument vs. algorithmic errors. Finally, there is a lack of explanation on how exactly this leads in to AOS and PACE. Since both missions will have ==higher resolution sensors== compared to POLDER and the abstract makes mention of these missions, a more detailed discussion on how this work might translate is warranted.

These are good points. The point on important factors, we hope, has been addressed by the addition of figures 7, 8, 9, 10, 11, B1, and D1. For solar zenith, see the figures in the response to RC1, as well as appendix D. We found no strong relationship between solar geometry and model skill. As for cloud types, (new) figures 7, 8, 9, and 10 analyze the role of cloud horizontal and vertical extent.

We added lines 476-479 to discuss the relationship between this work and the higher resolution sensors on PACE and AOS.

Technical Comments:

1. Line 78: 3DeepCT does not perform a segmentation task but rather a regression.
   Thank you, this has been corrected.

2. Line 125: The term "test set" might be unfamiliar to those not working with machine learning. It is recommended that the authors add a sentence stating that the test set is not seen by the model during training or validation, and is, therefore, the true test of the model's performance.
   Lines 131-135 contain an explanation of the training / validation / testing set conventions in machine learning. "Neither parameters nor hyperparameters should be optimized with respect to the test set, which is the final measure of a method's efficacy."

3. Line 277: What experiments is this referring to? A sentence prior to this should be added to clarify as it makes subsequent text harder to understand.
   The same training procedure (loss, optimizer, number of epochs, etc) is used for all experiments.

4. Line 281: No explanation is provided as to why the model performed worse with data augmentation. Could the authors elaborate and if so, add that to the text? Even if the exact reason is not known the readers and the applied machine learning community would appreciate more specifics on why/why not.
You raise a good point. We added a hypothesis for why data augmentation worsened performance (lines 306-309).

5. Line 288: Again, the term "binary cross-entropy loss" would benefit from a brief explanation as to how it optimizes the network better than other loss functions.
We added an explanation (lines 300-303).

6. Line 293 and line 315: Which tables and figures? The reader should not be expected to hunt for the relevant table and figures
This was an oversight and has been corrected, thank you.

7. Line 294: The first two sentences of this paragraph should be moved up to the start of section 3.4 where it is relevant and needed. Then, this should be recapped in this section to transition to the thresholding.
The model's output is treated slightly differently when used by the loss function and the evaluation function. We intentionally separated these so that the loss function is described in the approach section and the evaluation function is described in the results section, where it is immediately relevant.

**Citation**: https://doi.org/10.5194/egusphere-2023-2392-RC2

---

## Referee Report (RR1)

Egusphere-2023-2392-v1

General Comments:

The revised version addresses the major comments raised by the reviewers. I can recommend acceptance of the manuscript after minor revisions to address the remaining points discussed below and other technical comments.

**Specific Comments:**

The conclusions lack a summary of the quantitative results of the paper and also lack a recommendation about the types of clouds that this technique should be applied to by a potential user.

The results presented on accuracy by extent/altitude should be combined with studies from the literature on cloud types and their sizes (e.g., Wood & Field, 2011) to come to such a recommendation.

Wood, R., and P. R. Field, 2011: The Distribution of Cloud Horizontal Sizes. J. Climate, 24, 4800–4816, https://doi.org/10.1175/2011JCLI4056.1.

Figure 8: Is there a joint dependence of the dice score on the Top-of-clouddistance/Cumuluative Depth on the total thickness of the cloud object (highest-top to lowest base)? In other words, is there a difference between the accuracy at the top of a thick cloud and the top of a thin cloud? Alternatively, is there a difference between the top of a high-topped cloud and the top of a low-topped cloud?

Line 13: The abstract says "we draw conclusions" but doesn't state the conclusions. The abstract is most helpful when it summarizes these conclusions. I would recommend trying to trim words from the first half of the abstract and adding some more quantitative results to the abstract.

Technical Comments:

Line 21: References for the feedback cycles should be added here.

Line 27: Three satellite missions. There should be a transition sentence introducing satellite remote sensing as a means of acquiring semi-global observations to reduce these uncertainties.

Line 49: Some references to instruments like MISR/ATSR here would be good.

Line 60: I suggest changing this to something along the lines of "This work is the first to utilize POLDER measurements for the estimation of full vertical cloud profiles." So that readers also learn about what you are doing at the same time as you mention its novelty. This is important as the introduction currently lacks a clear "we are going to do X" statement. Line 62: The way the pinhole vs rational polynomial comparison is made in the text makes it seem like this is an algorithmic choice, rather than something that I would believe would originate from differences in optical hardware.

Line 332: I suggest adding "To evaluate the predictions of the models we use the Dice score." Or something to this effect at the beginning of Section 4.1.

Line 341: Here I suggest simply stating: we report the dice scores in %.

Line 358-359: Dice scores need percentage symbol.

Table 3/Section 4.4. which model architecture is used to produce these results? This should be in the caption of Table 3 and Figure 4.

Figure 9: I believe the color label should show something like "cloud length" rather than "color scale (km)". The color map for the clouds should be switched to something that doesn't end in black (e.g. red, yellow, blue) so that all the clouds can still be seen regardless of whether they are small.

---

## Author Response (AR2)

Dear reviewer,

We'd like to thank you for the insight you provided on our paper. Your suggestions have been included, and the revisions document shows all changes.

**Specific Comments:**

> The conclusions lack a summary of the quantitative results of the paper and also lack a recommendation about the types of clouds that this technique should be applied to by a potential user.

The conclusion has been amended to include a discussion of our results and recommendations.

> The results presented on accuracy by extent/altitude should be combined with studies from the literature on cloud types and their sizes (e.g., Wood & Field, 2011) to come to such a recommendation.

The suggested citation has been included, and we mention that it agrees with our findings.

> Figure 8: Is there a joint dependence of the dice score on the Top-of-cloud-distance/Cumulative Depth on the total thickness of the cloud object (highest-top to lowest base)? In other words, is there a difference between the accuracy at the top of a thick cloud and the top of a thin cloud? Alternatively, is there a difference between the top of a high-topped cloud and the top of a low-topped cloud?

We slightly clarified the wording in the text (line 427). Results at top of cloud are generally good across our experiments, although the model performs worse with quite thin clouds, which can be seen in the qualitative results. The altitude figures already illustrate the model's skill at high-topped vs low-topped. We choose not to add more figures relating cloud thickness to top-of-cloud distance and cumulative depth (as well as the correlations between all these variables) as the existing relationship is already quite complex, and because we believe Figures 7-8 are a better representation the model's performance for different cloud types/regions.

> Line 13: The abstract says "we draw conclusions" but doesn't state the conclusions. The abstract is most helpful when it summarizes these conclusions. I would recommend trying to trim words from the first half of the abstract and adding some more quantitative results to the abstract.

Good point. The abstract has been significantly changed to take this advice into account.

**Technical Comments:**

Line 21: References for the feedback cycles should be added here.

Added a citation for feedbacks, but also note that the citations already here relate to *both* cloud feedbacks and climate sensitivity models, which is why they're listed at the end.

Line 27: Three satellite missions. There should be a transition sentence introducing satellite remote sensing as a means of acquiring semi-global observations to reduce these uncertainties.

Fixed.

Line 49: Some references to instruments like MISR/ATSR here would be good.

MISR and ATSR are both already referenced, here or in the next paragraph.

Line 60: I suggest changing this to something along the lines of "This work is the first to utilize POLDER measurements for the estimation of full vertical cloud profiles." So that readers also learn about what you are doing at the same time as you mention its novelty. This is important as the introduction currently lacks a clear "we are going to do X" statement.

Good point, done!

Line 62: The way the pinhole vs rational polynomial comparison is made in the text makes it seem like this is an algorithmic choice, rather than something that I would believe would originate from differences in optical hardware.

This has been clarified.

Line 332: I suggest adding "To evaluate the predictions of the models we use the Dice score." Or something to this effect at the beginning of Section 4.1.

Fixed.

Line 341: Here I suggest simply stating: we report the dice scores in %. Line 358-359: Dice scores need percentage symbol.

Fixed.

Table 3/Section 4.4. which model architecture is used to produce these results? This should be in the caption of Table 3 and Figure 4.

Fixed.

Figure 9: I believe the color label should show something like "cloud length" rather than "color scale (km)". The color map for the clouds should be switched to something that doesn't end in black (e.g. red, yellow, blue) so that all the clouds can still be seen regardless of whether they are small.

Good point about the color map, it's been fixed now. We left the "color scale" label because the plot title already reads "Cloud Horizontal Extent". Adding it again would be redundant and calling it "cloud length" might mislead readers into thinking those are different things.

---

## Author Response (AR3)

We thank the editors and reviewers for their comments. Specific comments are addressed below:

I just noticed that your figures 2 and A1 contain satellite or airborne images. If you are not the originator of the images, then appropriate credit or copyright must be given. If applicable, please add the necessary details to the figure or the figure caption. Please make sure that the figure or caption contains the appropriate image credit as this is the responsibility of the authors.

We made these figures and the images within ourselves. We clarified that the images are derived from POLDER-3 in the captions, but the images were produced by original code written by the authors.

Dear authors, thank you for making changes based on the reviewer's recommendation. The reviewer had asked for a more quantitative assessment in the conclusions, and I still do not quite see that. A qualitative addition has been made. Could you please take another look and try to make things more quantitative (use some quality metrics instead of "good agreement")? Otherwise the conclusions will remain a bit vague. Please make sure that you did in fact address the reviewer's recommendations thoroughly, or state why you did not do that.

We added specific numbers to support the statements in the conclusion about the quantitative results.

---

## Author Response (AR4)

We thank the editors for their revisions and ongoing work to improve our paper. Specific comments are addressed below:

"Although the abstract was revised, quantitative information still needs to be added (at the very least, include a concise summary of the take home message or results). Examples of missing information: Dependence of accuracy on cloud size is a more significant signal than the view angle or spectral channels but that seems to be absent from the abstract. The Dice score could be used in the abstract without explaining it there ('high values are better') if that helps."

We added more about the quantitative results to the abstract:
- More angles -> higher skill
- Oxygen-A band -> strongly influences skill
- Multilayer clouds, horizontally small clouds, and low clouds over land -> low skill
- Tall clouds -> surprisingly high skill

We did not add actual numbers to the abstract, as we do not expect most readers to be familiar with the Dice score and feel that the metric should be introduced before reporting numbers. We also do not want to report the accuracy numbers, as they are not very informative about the model's skill (as discussed in the paper). Our hope is that readers will find the tables and figures in the paper and appendices to be the most informative (and easiest to read) formats of our quantitative results.

"In the conclusions, they did add quantitative results for the angle /spectral coverage. I would also prefer to see other results similarly summarized, for example, the land/sea contrast in accuracy."

We summarize the land/sea contrast in accuracy as well as the takeaways from the vertical/horizontal extent analysis previously requested by the reviewers.

"In summary, I recommend to go through the manuscript one more time and emphasize the story line and take home message. That can happen through more quantitative language as recommend by the reviewer, but possibly also other ways of emphasis."

Our additions to the abstract and conclusion should help more completely support the take home message in the paper.

---

## Author Response (AR5)

Dear reviewers,

We thank you all for your hard work on this paper.

The only changes made at this stage were to the README, which is part of the code repository associated with this manuscript. The changes to the README should be updated on SeaBASS the same day as this response (9/23/2024).

For clarity, we have included the revised text of the relevant section of the README below:

**Download and Unzip Code**

1. If you haven't already, then register for an EarthData account: https://urs.earthdata.nasa.gov/home
2. Go to SeaBASS search: https://seabass.gsfc.nasa.gov/search Scroll down to Keyword Search Filters. In the Search String box, type "ATCS". Scroll to the bottom and click Search.
3. Either check the box for "Include all associated files" and then click "Place Order All", or you can click the "Associated Files Info" tab to download files individually. In either case, place all the files into a single directory. You'll notice the code is in an archive named "41720ff87d_atcs_code_associated.tgz".
4. Unzip the code with the following command:

    tar -xzf <path_to_directory>/atcs_code_associated.tgz <path_to_where_you_want_the_code>